# Outlier Suppression: Pushing the Limit of Low-bit Transformer Language Models

**Xiuying Wei[1,2] , Yunchen Zhang[2,4] , Xiangguo Zhang[2] , Ruihao Gong[1,2],**
**Shanghang Zhang[3] , Qi Zhang[2] , Fengwei Yu[2] , Xianglong Liu[1]***

[1]State Key Lab of Software Development Environment, Beihang University
[2]SenseTime Research, [3]Peking University
[4]University of Electronic Science and Technology of China
{weixiuying, zhangyunchen, zhangxiangguo, gongruihao}@sensetime.com
shanghang@pku.edu.cn, xlliu@buaa.edu.cn

## Abstract

Transformer architecture has become the fundamental element of the widespread natural language processing (NLP) models. With the trends of large NLP models, the increasing memory and computation costs hinder their efficient deployment on resource-limited devices. Therefore, transformer quantization attracts wide research interest. Recent work recognizes that structured outliers are the critical bottleneck for quantization performance. However, their proposed methods increase the computation overhead and still leave the outliers there. To fundamentally address this problem, this paper delves into the inherent inducement and importance of the outliers. We discover that $\gamma$ in LayerNorm (LN) acts as a sinful amplifier for the outliers, and the importance of outliers varies greatly where some outliers provided by a few tokens cover a large area but can be clipped sharply without negative impacts. Motivated by these findings, we propose an outlier suppression framework including two components: Gamma Migration and Token-Wise Clipping. The Gamma Migration migrates the outlier amplifier to subsequent modules in an equivalent transformation, contributing to a more quantization-friendly model without any extra burden. The Token-Wise Clipping takes advantage of the large variance of token range and designs a token-wise coarse-to-fine pipeline, obtaining a clipping range with minimal final quantization loss in an efficient way. This framework effectively suppresses the outliers and can be used in a plug-and-play mode. Extensive experiments prove that our framework surpasses the existing works and, for the first time, pushes the 6-bit post-training BERT quantization to the full-precision (FP) level. Our code is available at https://github.com/wimh966/outlier_suppression.

## 1 Introduction

Transformer [1] has been one of the most common architectures in natural language processing along with lots of popular self-supervised models, such as BERT [2], RoBERTa [3], XLNet [4] and BART [5]. While these pre-trained models have demonstrated a significant superiority in performance, the memory and computation overheads have been a popular concern, particularly in the real development. Therefore, model compression [6, 7, 8, 9] has attracted much attention from both academia and industry. Among them, quantization [10, 11, 12, 13, 14, 15, 16, 17, 18, 19, 20], working in the low-precision arithmetic fashion, is one of the key approaches to compress large models and fit them into the lightweight devices.

---

*Corresponding author.

36th Conference on Neural Information Processing Systems (NeurIPS 2022).

These days, researchers focus more on quantization of Transformer-based models. [21] proposes an 8-bit quantization scheme for BERT-like models. [22] advises a group-wise quantization technique and analyzes mixed-precision using second-order Hessian information. [23, 24] combine distillation with quantization. [25] approximates nonlinear operations to implement integer-only quantization. Nonetheless, few studies investigate the inherent bottleneck of quantizing Transformer-based models.

Recently, some papers [26, 27] indicate that the Transformer-based models hold significantly large outliers (even close to 100) and these extreme outliers behave in structured patterns (mainly gather at a few embedding dimensions and even become larger on unique tokens). These special outliers can bring devastating damage to the quantization performance (e.g., a 12% drop even for the 8-bit [26]). To combat this challenge, existing method [26] chooses bypassing solutions such as a finer quantization granularity. However, this scheme causes an increased computation cost and unavoidably hinders the acceleration effect.

In this paper, to suppress the outliers rather than walk around them, we make an in-depth analysis to investigate the inducement of the outliers and the impact of clipping the outliers. For the inducement, we find that the scaling parameter $\gamma$ in the LayerNorm structure works as an outlier amplifier, which amplifies the outliers in the output. By extracting it, the activation becomes more robust for quantization. Then by further studying the clipping impact, we discover that the influence of final performance when clipping the outliers varies greatly, where some more aggressive outliers covering a large area can be clipped safely without accuracy degradation, but the accuracy can drop suddenly when the important outliers are clipped. More interestingly, though those less important outliers might present in a long tail form, they are only provided by a few tokens.

Motivated by the analysis, we propose an outlier suppression framework to push the limit of low-bit Transformer language models. Such framework contains two key components: Gamma Migration and Token-Wise Clipping, corresponding to the above two findings. The Gamma Migration produces a more quantization-friendly model by migrating the outlier amplifier $\gamma$ into subsequent modules in an equivalent transformation and bringing more robust activation for quantization without extra computation burden. The Token-Wise Clipping further efficiently finds a suitable clipping range with minimal final quantization loss in a coarse-to-fine procedure. The coarse-grained stage, which leverages the fact that those less important outliers only belong to a few tokens, can obtain a preliminary clipping range quickly in a token-wise manner. The fine-grained stage then optimizes it. Our proposed framework can be applied to different models and tasks, and coupled with existing methods. More essentially, the thought of outlier suppression shall shed new light on the study of NLP quantization.

To summarize, our contributions are as follows:

1. We delve into the inducement and clipping impact of outliers in the NLP models and draw two critical findings that help handle the bottleneck of transformer quantization.
2. Based on the findings, an outlier suppression framework containing Gamma Migration and Token-Wise Clipping is proposed. This framework is efficient, easy to implement, and plug-and-play.
3. The Gamma Migration suppresses the outliers from the inducement aspect and produces a more quantization-friendly model without any extra inference time. It transfers the outlier amplifier in LayerNorm to the subsequent modules in an equivalent transformation and contributes to activation with less quantization error.
4. The Token-Wise Clipping scheme suppresses the outliers from the aspect of importance and produces a superior clipping range efficiently. It can skip over those unimportant outliers quickly leveraging the large variance of token range and then focus on the influential area.
5. Extensive experiments on various NLP models (BERT, RoBERTa, BART) and tasks (text classification, question answering, and summarization) prove that our outlier suppression framework sets up a new state of the art for transformer quantization, and for the first time, pushes the 6-bit post-training quantization (PTQ) and 4-bit quantization-aware training (QAT) accuracy of BERT to the full-precision level.

## 2   Preliminaries

**Basic Notations.** We mark matrices as $\boldsymbol{X}$ and vectors as $\boldsymbol{x}$. Operator $\cdot$ denotes scalar multiplication, and $\odot$ is adopted for element-wise multiplication on matrices or vectors. Also, we use $\boldsymbol{W}\boldsymbol{x}$ as

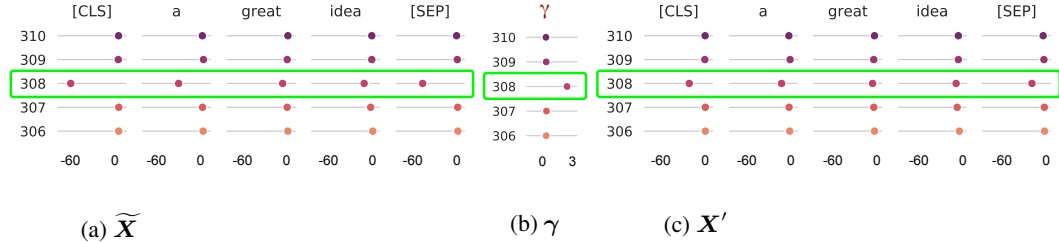

Figure 1: Presentation of outliers over $\widetilde{X}$, $\gamma$ and $X'$ of LayerNorm on BERT-SST-2. For example, at dimension 308, $\gamma$ and $\widetilde{X}$ both have sharper values. By excluding $\gamma$, it can be seen that $X'$ holds milder distribution than $\widetilde{X}$. More evidence is put in Sec. D.1.

matrix-vector multiplication. Specifically, considering the tokens in NLP tasks, $X_{t,j}$ stands for the element at token $t$ and embedding $j$, and $x_t$ represents the embedding of token $t$.

**Quantizer.** Quantization usually includes two operations.

$$\bar{x} = clip(\lfloor \frac{x}{s} \rceil + z, \, 0, \, 2^b - 1), \quad \hat{x} = (\bar{x} - z) \cdot s \tag{1}$$

where $s$ (step size), $z$ (zero point) are quantization parameters, $b$ is the bit setting. The first operation called "Quant" maps continuous numbers ($x$) to discrete points ($\bar{x}$) for integer-arithmetic-only matrix computation. The second operation called "DeQuant" recovers it to $\hat{x}$ after multiplication.

## 3 Outlier analysis

For Transformer-based models, standard 6/8-bit PTQ or 4-bit QAT would cause severe accuracy degradation. Investigating each quantizer, we recognize that the output of LayerNorm structures and GELU functions hold some sharp outliers, which should be responsible for the large quantization error. Evidence and experimental results in Sec. B.2.

To deeply investigate the relationship between the harmful outliers and quantization performance, we explore the underlying inducement and impact of clipping the outliers. Before that, some brief descriptions (see Sec. C.1 for detailed ones) about the outliers are given first to help understand the following two parts. The outliers show structured characteristics that they mainly gather at some certain embedding dimensions, and upon these dimensions, the outliers provided by unique tokens like the separate toke and comma even hold more aggressive values.

### 3.1 Inducement of outliers

For the inducement of outliers, we find that the scaling parameter in LayerNorm amplifies the outliers from embedding dimensions. And the phenomenon that some tokens have sharper outliers might be caused by the uneven token frequency in the pre-training phase (see Sec. C.2). In this part, we mainly explain the first inducement to solve these outliers from the origin. For another one, due to the high cost of adjusting the pre-training, we discuss the clipping impact in the next part to suppress these outliers from the clipping perspective.

Considering the challenges of quantizing the LayerNorm, the natural action is to dive into its internal structure. For token $t$ at $j^{th}$ embedding dimension, it first normalizes the input using mean ($u_t$) and variance ($\sigma_t^2$) each forward pass, then scales and shifts the value with parameter $\gamma_j$ and $\beta_j$.

$$\textbf{LayerNorm}: \quad \widetilde{X}_{t,j} = \frac{X_{t,j} - u_t}{\sqrt{\sigma_t^2 + \epsilon}} \cdot \gamma_j + \beta_j \tag{2}$$

Then, by observing the parameter distribution of LayerNorm, we surprisingly find that the multiplier $\gamma$ (Fig. 1b) and the output $\widetilde{X}$ (Fig. 1a) hold outliers at the same embedding dimensions. Besides, the adder $\beta$ denotes a smaller range (e.g., (0,3)) compared to the output range (e.g., (-60, 0)), so we

ignore it for identifying the key point. That is to say, $\gamma$ plays a crucial part for the outliers in Fig. 1a, especially can amplify the outliers across tokens by serving as a shared parameter.

This observation enlightens us to remove the amplification effect by extracting $\gamma$ from Eq. (2) and use the Non-scaling LayerNorm Eq. (3).

$$\mathbf{Non\text{-}scaling\ LayerNorm}:\quad \boldsymbol{X}'_{t,j} = \frac{\boldsymbol{X}_{t,j} - \boldsymbol{u}_t}{\sqrt{\boldsymbol{\sigma}_t^2 + \epsilon}} + \frac{\boldsymbol{\beta}_j}{\boldsymbol{\gamma}_j} \tag{3}$$

Fig. 1c and Fig. 1a show that the output of the Non-scaling LayerNorm denotes a milder distribution with weaker outliers than the normal one. It not only coincides with that $\gamma$ does strengthen the outliers but also reveals that $\boldsymbol{X}'$ behaves more friendly than $\widetilde{\boldsymbol{X}}$ for quantization.

To quantitatively validate the more quantization-friendly distribution $\boldsymbol{X}'$ holds, we adopt the cosine similarity metric to evaluate the quantization loss. From Table 1, the second row with higher similarity, namely less quantization error, explains that the quantization performance can be improved using Non-scaling LayerNorm.

| Tensor | 0 | 1 | 2 | 3 | 4 | 5 | 6 | 7 | 8 | 9 | 10 | 11 |
|---|---|---|---|---|---|---|---|---|---|---|---|---|
| $\widetilde{\boldsymbol{X}}$ | 97.16 | 97.03 | 97.61 | 94.37 | 93.41 | 93.53 | 93.31 | 93.61 | 94.56 | 95.62 | 96.13 | 98.57 |
| $\boldsymbol{X}'$ | 99.23 | 99.22 | 99.11 | 99.02 | 98.99 | 99.00 | 98.99 | 98.83 | 98.70 | 99.05 | 99.44 | 99.07 |

Table 1: Cosine similarity (%) of the quantized value (6-bit) and the real signal for $\widetilde{\boldsymbol{X}}$ and $\boldsymbol{X}'$ across 12 LayerNorm after Multi-Head Attention on BERT-SST-2. Higher is better. More evidence in Sec. D.1.

## 3.2 Impact of outlier clipping

In this part, we explore the impact of clipping the outliers to design a method that can find an appropriate clipping range for quantization. The experiments are designed for the clipping impact on the accuracy and token of FP models.

**Impact on accuracy.** When clipping the outliers and evaluating the final performance, we find that the importance of outliers is highly varied. Take the outliers after GELU as an example here (others in Sec. D.2), Fig. 2 shows that clipping the more aggressive outliers sharply (clipping signals in 10-100 to 10) even does not hurt the full-precision performance with accuracy still at 91.02, while the accuracy drops suddenly to 85.93 with too many outliers cut.

**Impact on token.** Another key point is the unimportant outliers which can be clipped without even any accuracy drop in FP models only correspond to a few tokens. Motivated by [26], they refer that the separator token [SEP] attends to larger values. We are also aware of the different ranges provided by different tokens. From the red points in Fig. 2, which represents the proportion of clipped tokens, it can be clearly seen that the more aggressive outliers though occupy a large range from 10 to 100 only matches with 3% tokens. Destroying those sharper outliers belonging to a few tokens will not affect the performance.

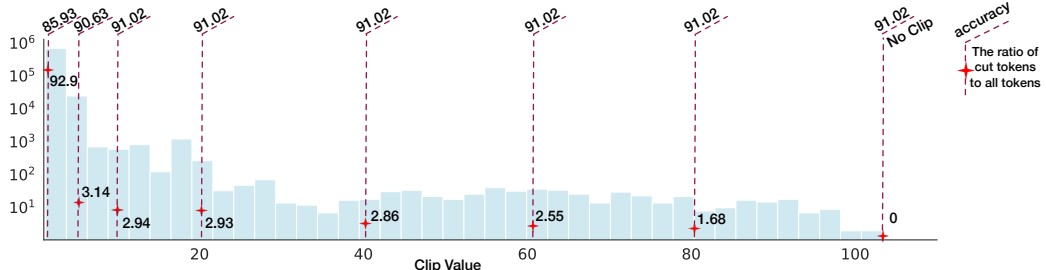

Figure 2: To detect the impact of clipping the outliers, we first draw the distribution using (mean + 3 * std) as its left border, then enumerate the value to cut the tensor on RoBERTa-QNLI. Red points reflect the proportion of clipped tokens. More evidence in Sec. D.2.

The former investigation of accuracy impact suggests us taking the final performance into account to find a superior clipping range, where some local optimization methods like [28] are not suitable here. The latter finding in token impact encourages us to leverage the token's indication to quickly skip over the unimportant area, especially when it presents in a long tail form where some methods like [29] suffer low efficiency. Based on these, we will introduce our method in Sec. 4.2.

# 4 Method

In this section, we propose our outlier suppression framework based on the above analysis. Firstly, the Gamma Migration technique is adopted to obtain a more quantization-friendly model by migrating the gamma into subsequent modules. Secondly, the Token-Wise Clipping further finds a suitable clipping range efficiently by leveraging the large variance of the token range.

## 4.1 Gamma Migration

As pointed out in Sec. 3.1, activation without going through the scaling parameter provides less quantization error. In this way, we split the LayerNorm function, migrate $\gamma$ into follow-up structures and quantize the output of the Non-scaling LayerNorm. The transformation is equivalent for the FP model and brings more robust activation for the low-bit one. The overall flow is illustrated in Fig. 3.

**Migration equivalence on FP model.** Naturally, as referred in Eq. (3), we extract the parameter $\gamma$ and transform the LayerNorm into Non-scaling one, thus seperate $X'_{t,j}$ from $\widetilde{X}_{t,j}$.

$$\widetilde{X}_{t,j} = X'_{t,j} \cdot \gamma_j \tag{4}$$

Since the residual connection is frequently adopted after LayerNorm ([30, 31, 32]), it is necessary to illustrate the way to migrate parameter $\gamma$ into two branches. To be specific, considering the LayerNorm after Multi-Head Attention (Fig. 3), $\gamma$ will be excluded from the LayerNorm and moved to the shortcut branch and weight of the next layer. Then the LayerNorm becomes the Non-scaling one, the shortcut branch establishes a new parameter $\gamma$, and the weight of the next layer can absorb the $\gamma$.

Now, we show how the weight absorbs $\gamma$. For linear layers, we have the following equation:

$$W(x \odot \begin{bmatrix} \gamma_1 \\ \gamma_2 \\ ... \\ \gamma_n \end{bmatrix}) = (W \odot \begin{bmatrix} \gamma_1 & \gamma_2 & ... & \gamma_n \\ \gamma_1 & \gamma_2 & ... & \gamma_n \\ ... & & & \\ \gamma_1 & \gamma_2 & ... & \gamma_n \end{bmatrix})x, \tag{5}$$

where $x$ serves as a column vector and $\gamma \in \mathbb{R}^n$. The proof is available in Appendix A. Because $\gamma$ is a shared parameter, each token's embedding satisfies Eq. (5), which promises success of transferring the $\gamma$ into the next layer's weight.

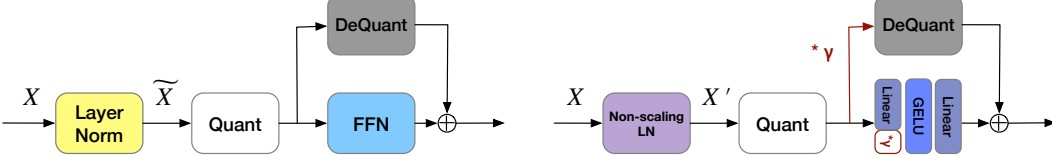

Figure 3: Comparison of the quantization flow before (left) and after (right) Gamma Migration. The original LayerNorm = the Non-scaling LayerNorm * $\gamma$. For other detailed applications such as LayerNorm in encoder-decoder structure, see Fig. 6, Fig. 7.

**Quantization after migration.** Deriving from the above equivalent transformation, we outline the quantization pattern after the migration process. From Fig. 3, the "Quant" process is employed at $X'$, then the quantized output engages in the matrix multiplication on one branch, multiplies parameter $\gamma$ and experiences the "DeQuant" process on another branch. In fact, this means delaying the $\gamma$ calculation from LayerNorm to the shortcut branch. Hence, this new design will not increase the computation overhead.

**Effect of migration.** We then analyze the effect of Gamma Migration on weight and activation, respectively, to reveal that the activation quantization burden has been greatly alleviated with relatively

a slight influence on weight. To begin with, suppose that the absolute max range of output in the original LayerNorm is $|max(\boldsymbol{X}')| * |max(\boldsymbol{\gamma})|$ for the reason that outliers emerge at the same embedding dimensions among $\boldsymbol{\gamma}$, activation before $\boldsymbol{X}'$ and after $\widetilde{\boldsymbol{X}}$ scaling function. For activation, extracting the $\boldsymbol{\gamma}$ will reduce the activation range by $|max(\boldsymbol{\gamma})|$ times. And the results in Table 1 have already validated the profit the transformation brings to activation. For weight, the weight matrix does not have the same embedding outlier phenomenon as the activation. Therefore, the weight range will not be amplified $|max(\boldsymbol{\gamma})|$ times after the migration. Experimentally, we also calculate the cosine similarity for the changed weight and observe that $\boldsymbol{\gamma}$ has little impact on weight (Table 2).

| Tensor | 0 | 1 | 2 | 3 | 4 | 5 | 6 | 7 | 8 | 9 | 10 | 11 |
|---|---|---|---|---|---|---|---|---|---|---|---|---|
| original weight | 99.95 | 99.95 | 99.95 | 99.95 | 99.95 | 99.95 | 99.95 | 99.95 | 99.95 | 99.95 | 99.95 | 99.95 |
| changed weight | 99.95 | 99.95 | 99.95 | 99.90 | 99.90 | 99.92 | 99.94 | 99.95 | 99.95 | 99.95 | 99.91 | 99.94 |

Table 2: Cosine similarity (%) between the quantized value (6-bit) and the real signal for original weight and the changed weight across 12 Intermediate layers on BERT-SST-2. It can be seen that there is little disparity between the two rows, especially compared with Table 1.

## 4.2 Token-Wise Clipping

Based on the analysis, we propose the Token-Wise Clipping method which considers the final loss when finding a clipping range and takes a coarse-to-fine paradigm to minimize it efficiently in a token-wise manner.

Regarding the very different accuracy impact of clipping the outliers, we search the clipping range, equivalently the step size $s$, which has the minimal distance between the final quantized output $\hat{f}(s)$ and the real one $f$ defined as Eq. (6). To implement the process efficiently, especially when the unimportant outliers cover a wide area, a coarse-to-fine paradigm is designed below.

$$L(s) = \|\hat{f}(s) - f\|_F^2, \tag{6}$$

**Coarse-grained Stage.** At this stage, our aim is to quickly skip over the area where clipping causes little accuracy influence. According to Sec. 3.2, the long tail area only matches with a few tokens. Therefore, we suggest using the max value of the embedding at token $t$ to be its representatives (min value as representatives for negative outliers). A new tensor with $T$ elements can be constructed by taking out the maximum signal for each token:

$$\boldsymbol{o}^u = \{max(\boldsymbol{x}_1), max(\boldsymbol{x}_2), ..., max(\boldsymbol{x}_T)\}, \tag{7}$$

where $\boldsymbol{o}^u$ is marked as the collection of upper bounds, $\boldsymbol{o}^l$ as the collection of lower bounds.

Then for a clipping ratio $\alpha$ on $\boldsymbol{o}^u$, calculate the corresponding clipping value $c^u$ and use it to cut the tensor.

$$c^u = quantile(\boldsymbol{o}^u, \alpha), \tag{8}$$

where the quantile function computes the $\alpha\text{-}th$ quantiles of $\boldsymbol{o}^u$.

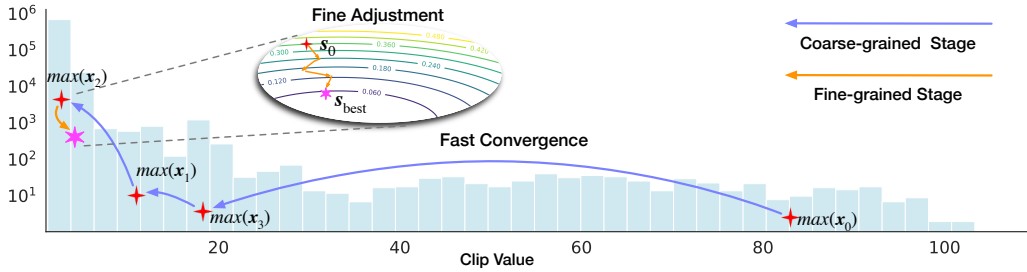

Figure 4: Flow diagram of the proposed Token-Wise Clipping

Through grid search of token-wise clipping ratio, step size $s = \frac{c^u - c^l}{2^b - 1}$ ($b$ is the bit-width) with minimal quantization loss Eq. (6) is obtained. We mark it as $s_0$ for later optimization.

**Fine-grained Stage.** At this stage, our aim is to make some fine-grained adjustments in the critical area to further provide a guarantee for the final effect. In detail, with the initialization $s_0$, a learning procedure based on gradient descent is used to update parameter $s$ towards loss $L(s)$ with learning rate $\eta$, as described in Eq. (9).

$$s = s - \eta \frac{\partial L(s)}{\partial s} \tag{9}$$

**Benefits.** We mainly explain the benefits of the coarse-grained stage here from efficiency and quantization performance, where the experimental comparisons with other existing approaches are put in Sec. D.3. For efficiency, because the wide range of outliers only corresponds to a few tokens, passing through the unimportant area from the token perspective needs much fewer iterations than from the value perspective. Moreover, the representative collection reduces the size of the tensor ($o^u$ distilled from $X$), so the method can run very fast each iteration. For quantization performance, the first coarse step has already produced a suitable clipping range (Sec. 5.2), which offers a good initialization point for upcoming tuning.

## 5 Experiments

In this section, we conduct two sets of experiments to verify the effectiveness of our outlier suppression framework. Sec. 5.2 shows the effect of each component. Sec. 5.3 lists the results compared with other existing approaches across text classification, question answering, and summarization tasks. On the whole, we evaluate GLUE benchmark [33], SQuAD [34, 35], and XSum [36] and CNN/DailyMail [37] across BERT, RoBERTa, and BART models. Here, 4-4-4 presents 4-bit weight, embedding, and activation. And the model size under a certain bit is put in Table 17.

### 5.1 Experimental setup

**Implementation details.** To begin with, we identify the quantization nodes and take a reasonable scheme like the one in FasterTransformer [38] (Details see Sec. B.1). For PTQ, equipping our framework, we use 256 samples to calibrate the model. For QAT, our methods work on the calibration phase and later are combined with LSQ+ [12], a strong baseline for the training phase. For training, hyper-parameters like learning rate are searched both for our methods and baseline techniques for fair comparisons. Details see Appendix F.

**Baseline.** For PTQ, we compare with the prevalent calibration mechanisms including MinMax [39], OMSE [28], Percentile [29], EasyQuant [40] and PEG [26]. For QAT, we present the results of Q-BERT [22], Q8BERT [21] and PEG [26]. Also, because our framework applying in QAT is coupled with LSQ+ [12], we show the results of the pure LSQ+, and another canonical quantization approach PACT [41]. Last but not least, the results combined with knowledge distillation (KD) proposed in TernaryBERT [23] are included as well.

### 5.2 Ablation Study

In this subsection, we ablate the design elements in the proposed framework (Table 3). As a general plug-in module, Gamma Migration helps both the MinMax and Token-Wise Clipping. And the Token-Wise Clipping also surpasses the baseline by a large margin: 17.53% on QNLI, 13.22% on MRPC (comparisons with other calibration algorithms see Sec. D.3). About the phenomenon that the fine-grained stage sometimes does not improve much upon the coarse-grained one, we think it's due to the already good enough results produced by the coarse step.

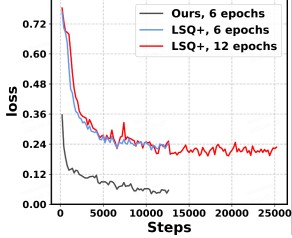

Figure 5: QAT fine-tuning process on BERT-SST-2.

Besides, Fig. 5 conveys that with a good initialization point provided by our framework, the training of QAT becomes much faster and easier.

| Method | CoLA (Matt.) | MNLI (acc m/mm) | MRPC (f1/acc) | QNLI (acc) | QQP (f1/acc) | RTE (acc) | SST-2 (acc) | STS-B (Pear./Spear.) |
|---|---|---|---|---|---|---|---|---|
| FP32 | 62.50 | 87.75/87.23 | 93.1/90.44 | 92.68 | 88.78/91.6 | 80.51 | 95.18 | 91.04/90.72 |
| Baseline (MinMax) | 0.0 | 34.9/35.0 | 71.64/67.4 | 62.13 | 51.88/74.37 | 49.82 | 77.87 | 44.11/46.74 |
| MinMax + Gamma Migration | 0.0 | 53.53/54.64 | **87.97/82.84** | 78.56 | 78.04/85.3 | 55.6 | 85.67 | 61.03/63.22 |
| Token-Wise Clipping (Coarse) | 34.95 | 80.56/80.84 | 85.05/79.41 | 79.46 | 85.96/89.31 | 66.43 | 91.63 | 82.03/82.45 |
| Token-Wise Clipping | 37.64 | 81.13/81.26 | 85.59/79.9 | 79.66 | 85.83/89.26 | 64.62 | 91.63 | 83.10/83.51 |
| Gamma Migration + Token-Wise Clipping | **46.35** | **83.38/83.32** | 87.50/83.33 | **86.82** | **86.82/90.01** | 67.51 | **92.2** | **86.83/86.93** |

Table 3: Results of our proposed Gamma Migration and Token-Wise Clipping for RoBERTa with 6-bit PTQ.

## 5.3 Main Results

### 5.3.1 Results on classification tasks

| Method | Bits (W-E-A) | CoLA (Matt.) | MNLI (acc m/mm) | MRPC (f1/acc) | QNLI (acc) | QQP (f1/acc) | RTE (acc) | SST-2 (acc) | STS-B (Pear./Spear.) | Avg. |
|---|---|---|---|---|---|---|---|---|---|---|
| BERT | 32-32-32 | 59.60 | 84.94/84.76 | 91.35/87.75 | 91.84 | 87.82/90.91 | 72.56 | 93.35 | 89.70/89.28 | 83.83 |
| MinMax | 8-8-8 | 57.08 | 82.77/83.47 | 89.90/85.78 | 90.76 | 87.84/90.74 | 69.68 | 92.78 | 86.83/88.56 | 82.28 |
| OMSE [28] | 8-8-8 | 57.15 | 84.04/84.29 | 90.10/85.78 | 91.12 | 87.64/90.54 | 72.20 | 93.23 | 87.90/88.65 | 82.90 |
| **Ours** | 8-8-8 | **61.64** | **84.38/84.53** | **91.44/87.75** | **91.49** | **87.92/90.77** | **72.20** | **93.81** | **89.23/89.01** | **83.96** |
| OMSE | 6-6-6 | 35.44 | 74.00/73.30 | 81.54/76.47 | 84.66 | 76.07/82.12 | 64.26 | 86.27 | 85.57/86.05 | 73.52 |
| Percentile [29] | 6-6-6 | 37.32 | 72.40/71.69 | 85.09/79.90 | 79.37 | 72.58/80.19 | 61.73 | 87.27 | 86.38/87.29 | 72.93 |
| EasyQuant [40] | 6-6-6 | 38.16 | 75.82/75.66 | 82.51/77.45 | 84.94 | 75.31/81.81 | 65.34 | 87.27 | 85.50/86.33 | 74.49 |
| **Ours** | 6-6-6 | **54.40** | **82.02/81.69** | **87.45/83.33** | **89.82** | **84.69/88.94** | **70.76** | **91.86** | **88.65/88.55** | **81.19** |
| PEG [26] ♣ | 8-8-8 | 59.43 | 81.25 | 88.53 | 91.07 | 89.42 | 69.31 | 92.66 | 87.92 | 82.45 |
| **Ours** ♣ | 8-8-8 | **59.83** | **82.93/82.59** | **91.33/87.99** | 90.02 | 87.45/90.34 | **70.04** | 92.66 | **88.42/88.81** | **82.81** |
| PEG ♣ | 6-6-6 | 9.46 | 32.44/32.77 | 83.64/78.43 | 49.46 | 29.93/62.97 | 70.76 | 90.14 | 52.79/53.22 | 54.11 |
| **Ours** ♣ | 6-6-6 | **42.27** | **78.54/78.32** | **85.33/81.13** | **85.36** | **78.47/84.66** | 68.59 | **91.74** | **87.33/87.19** | **77.31** |
| RoBERTa | 32-32-32 | 62.50 | 87.75/87.23 | 93.1/90.44 | 92.68 | 88.78/91.6 | 80.51 | 95.18 | 91.04/90.72 | 86.40 |
| MinMax | 8-8-8 | 41.62 | 87.52/86.88 | 91.56/88.48 | 92.11 | 88.60/91.44 | 76.90 | 94.82 | 91.00/90.66 | 82.94 |
| OMSE | 8-8-8 | 38.59 | 87.32/87.14 | 92.39/89.46 | 92.51 | 87.95/90.95 | 76.53 | 94.61 | 90.95/90.65 | 82.58 |
| **Ours** | 8-8-8 | **62.50** | **87.61/87.31** | **92.39/89.46** | **92.53** | **88.64/91.49** | **78.34** | **94.95** | **91.08/90.73** | **85.96** |
| OMSE | 6-6-6 | 1.81 | 72.89/72.65 | 85.38/78.68 | 76.53 | 85.24/88.94 | 64.26 | 91.17 | 80.81/81.99 | 69.63 |
| Percentile | 6-6-6 | 20.73 | 72.23/73.68 | 84.83/78.43 | 77.16 | 82.21/87.44 | 62.82 | 88.19 | 79.41/79.64 | 70.98 |
| EasyQuant | 6-6-6 | 9.28 | 74.96/75.87 | 84.31/76.47 | 74.04 | 85.52/89.12 | 62.45 | 89.56 | 80.89/82.38 | 70.01 |
| **Ours** | 6-6-6 | **46.35** | **83.38/83.32** | **87.50/83.33** | **86.82** | **86.82/90.01** | 67.51 | **92.2** | **86.83/86.93** | **79.62** |
| BART | 32-32-32 | 56.32 | 86.45/86.55 | 91.37/87.50 | 92.31 | 88.34/91.39 | 79.06 | 93.35 | 90.11/89.94 | 84.61 |
| MinMax | 8-8-8 | 55.38 | 85.87/86.14 | 89.44/85.29 | 91.20 | 88.07/91.24 | 77.98 | 93.69 | 89.90/89.73 | 83.89 |
| OMSE | 8-8-8 | 54.56 | 85.6/86.25 | 90.31/86.27 | 90.74 | 88.21/91.3 | 78.7 | 93.58 | 90.07/89.88 | 83.94 |
| **Ours** | 8-8-8 | **55.53** | **86.28/86.17** | **90.40/86.52** | **91.47** | **88.25/91.35** | **80.51** | **93.92** | **90.20/89.95** | **84.50** |
| OMSE | 6-6-6 | 31.06 | 41.92/42.08 | 56.37/54.36 | 52.72 | 78.96/86.02 | 51.99 | 87.39 | 84.38/85.69 | 61.01 |
| Percentile | 6-6-6 | 26.21 | 74.72/75.29 | 83.52/74.26 | 53.71 | 82.64/87.48 | 67.15 | 87.96 | 63.99/65.01 | 67.31 |
| EasyQuant | 6-6-6 | 25.66 | 43.48/43.27 | 59.26/59.56 | 50.76 | 81.89/87.67 | 52.71 | 87.73 | 85.39/86.74 | 61.31 |
| **Ours** | 6-6-6 | **44.51** | **82.46/82.98** | **86.41/80.88** | **86.34** | **83.60/88.45** | **71.12** | **90.94** | **87.56/87.38** | **79.10** |

Table 4: PTQ performance on GLUE benchmark. ♣: results taking the same quantization nodes with PEG [26] for fair comparisons. For the percentile, we search the hyper-parameter in [0.999, 0.9999, 0.99999] and report the best on dev set.

**PTQ.** Table 4 shows the results of PTQ on GLUE tasks. For 8-bit BERT models, although previous methods generally behave well, our methods can still achieve satisfying outcomes even on small datasets such as CoLA (4.49% upswings) and STS-B (1.33% upswings). To fully exploit the limit, we try a more inspiring setting with weight and activation quantized to 6-bit. It can be seen that ours is indeed close to FP value within 2.64% overall. Meanwhile, we also compare with PEG [26] fairly by taking their quantization nodes. To be noted, their per-embedding-group (PEG) quantization certainly brings extra computation overhead and might not be available on real deployment while ours brings favorable results and can enjoy lossless acceleration on hardware. Besides, the experimental results on RoBERTa and BART consistently demonstrate our superiority whereas existing methods suffer from a non-negligible accuracy drop. On average, ours achieves up to 8.64% and 11.79% better accuracy on RoBERT and BART. To conclude, our proposed methods push the limit of 6-bit quantization to a new state of the art.

**QAT.** In particular, we prove the compatibility of our methods on QAT. Table 5 lists the results on BERT, other see Sec. D.4. In a much harder setting (4-4-4 bit quantization), our outlier suppression framework wins near-floating-point performance with a reduction of 2.70% on average on 4-bit quantization. Yielding a good initialization, ours obtain an acceptable accuracy drop (0.7% on QQP, 1.7% on MNLI) without any distillation and data augmentation trick, versus 4.19% and 3.16% of LSQ+. Furthermore, ours still enables performance improvements working with knowledge distillation, especially at 2-bit weight and embedding.

| Method | Bits (W-E-A) | CoLA (Matt.) | MNLI (acc m/mm) | MRPC (f1/acc) | QNLI (acc) | QQP (f1/acc) | RTE (acc) | SST-2 (acc) | STS-B (Pear./Spear.) | Avg. |
|---|---|---|---|---|---|---|---|---|---|---|
| BERT | 32-32-32 | 59.60 | 84.94/84.76 | 91.35/87.75 | 91.84 | 87.82/90.91 | 72.56 | 93.35 | 89.70/89.28 | 83.83 |
| Q8BERT [21] | 8-8-8 | 58.48 | - | 89.56/- | 90.62 | 87.96/- | 68.78 | 92.24 | 89.04/- | - |
| Q-BERT [22] | 8-4-8 | - | 78.08/78.96 | - | - | - | - | 85.55 | - | - |
| PACT [41] | 4-4-8 | 55.23 | 83.98/83.90 | 91.58/88.24 | 91.12 | 88.19/91.20 | 71.84 | 91.86 | 89.73/89.27 | 82.89 |
| LSQ+ [12] | 4-4-8 | 57.70 | 84.17/84.02 | 89.75/85.78 | 91.27 | 88.18/91.16 | 70.76 | 91.97 | **89.74/89.3** | 82.84 |
| PEG [26] | 4-4-8 | 57.42 | 84.22/84.52 | 89.90/85.78 | 90.46 | 88.15/91.25 | 67.87 | 92.78 | 89.36/88.95 | 82.45 |
| **Ours** | 4-4-8 | **61.06** | 84.82/84.89 | 91.26/87.75 | 91.41 | 88.45/91.40 | 73.65 | 92.55 | 89.71/89.24 | **84.05** |
| PEG | 4-4-4 | 0.0 | 35.45/35.22 | 81.22/68.38 | 49.46 | 0.0/63.18 | 52.71 | 76.26 | nan/nan | - |
| PACT | 4-4-4 | 0.0 | 74.17/74.85 | 84.97/80.15 | 87.31 | 81.68/86.14 | 62.09 | 83.03 | 81.64/81.43 | 69.37 |
| LSQ+ | 4-4-4 | 0.0 | 81.40/81.97 | 88.34/83.82 | 88.10 | 83.11/87.24 | 64.62 | 82.34 | 84.16/83.75 | 71.49 |
| **Ours** | 4-4-4 | **50.56** | 83.05/83.24 | 89.08/84.31 | 89.88 | 87.00/90.33 | 70.76 | 91.86 | 87.64/87.36 | **81.13** |
| PEG ♣ * | 4-4-8 | 57.22 | 83.69 | 87.77 | 91.29 | 89.64 | 70.04 | 92.32 | 89.13 | 82.64 |
| **Ours ♣** | 4-4-8 | **59.57** | 85.00/84.31 | 91.07/87.75 | 91.31 | 88.35/91.32 | 72.20 | 92.43 | 89.57/89.20 | **83.60** |
| PEG ♣ | 4-4-4 | 0.0 | 35.45/35.22 | 31.62/0.0 | 49.46 | 0.0/63.18 | 52.71 | 49.08 | -0.0219/-0.0199 | 29.25 |
| **Ours ♣** | 4-4-4 | **51.93** | 83.03/83.24 | 89.39/85.05 | 90.33 | 87.38/90.62 | 72.56 | 91.74 | 88.36/87.91 | **81.76** |
| LSQ+(+KD) | 4-4-4 | 14.98 | 83.59/84.06 | **92.47/89.46** | 91.16 | 87.96/91.01 | 67.87 | 85.55 | 84.17/83.96 | 75.99 |
| **Ours(+KD)** | 4-4-4 | **56.67** | 84.50/84.65 | 91.61/88.24 | 91.45 | 88.59/91.42 | 74.37 | 92.55 | 89.13/88.78 | **83.56** |
| LSQ+(+KD) | 2-2-4 | 9.44 | 83.45/83.38 | 88.03/82.60 | 90.66 | 87.1/90.36 | 55.60 | 83.60 | 36.69/35.89 | 66.63 |
| **Ours(+KD)** | 2-2-4 | **47.02** | 84.56/84.31 | 90.97/87.25 | 90.83 | 88.08/91.12 | 65.70 | 91.86 | 86.12/85.78 | 80.56 |

Table 5: Comparison among different QAT strategies with low-bit activation on GLUE benchmark for BERT. ♣: results taking the same quantization nodes with PEG [26] for fair comparisons. *: combined score for MNLI, MRPC, QQP and STS-B.

### 5.3.2 Results on question answering tasks

To demonstrate the wider applicability of our methods, we evaluate them on SQuAD datasets. When going down to 6-bit quantization, the performance of other methods drastically drops. Ours still outperforms them by over 4.73% and 15.55% on BERT and RoBERTa on SQuAD v1.1. Also, the boost can be 12.31% and 4.96% on RoBERTa and BART on SQuAD v2.0.

### 5.3.3 Results on summarization tasks

It is of high value to validate the effect of our methods on summarization tasks. We choose classical datasets CNN/DailyMail and XSum and report the ROUGE 1/2/L score of BART. Table 7 illustrates that our approaches also benefit the encoder-decoder models, and can bring a near-floating-point performance on 8-bit and about 4% enhancement on 6-bit.

| Method | Bits (W-E-A) | BERT | | RoBERTa | | BART | |
|---|---|---|---|---|---|---|---|
| | | SQuAD v1.1 | SQuAD v2.0 | SQuAD v1.1 | SQuAD v2.0 | SQuAD v1.1 | SQuAD v2.0 |
| Full Prec. | 32-32-32 | 88.28/80.82 | 77.34/73.60 | 92.25/85.83 | 83.30/80.26 | 91.63/84.79 | 80.82/77.41 |
| OMSE [28] | 8-8-8 | **87.90/80.16** | 76.88/73.08 | 91.48/84.53 | 82.53/79.41 | 90.49/83.11 | 79.62/76.12 |
| **Ours** | 8-8-8 | 87.60/79.80 | **76.93/73.14** | **91.57/84.86** | **82.94/79.72** | **91.08/84.07** | **80.55/77.04** |
| OMSE | 6-6-6 | 79.77/69.10 | 67.52/63.09 | 70.64/58.80 | 45.80/39.95 | 81.44/70.61 | 67.89/63.29 |
| Percentile [29] | 6-6-6 | 78.55/67.14 | 69.12/65.64 | 67.24/53.28 | 56.38/51.58 | 82.45/72.87 | 68.44/63.29 |
| EasyQuant [40] | 6-6-6 | 80.47/70.08 | 71.95/68.06 | 67.85/55.92 | 47.99/42.21 | 82.41/71.72 | 69.93/64.94 |
| **Ours** | 6-6-6 | **84.48/75.53** | **74.69/70.55** | **80.79/70.83** | **68.47/64.10** | **83.68/75.34** | **74.44/70.36** |

Table 6: Comparison among typical PTQ approaches in terms of f1/em on SQuAD.

| Method | Bits(W-E-A) | CNN DailyMail | XSum | Bits(W-E-A) | CNN DailyMail | XSum |
|---|---|---|---|---|---|---|
| Full prec. | 32-32-32 | 45.62/22.85/42.88 | 42.82/20.11/34.99 | 32-32-32 | 45.62/22.85/42.88 | 42.82/20.11/34.99 |
| OMSE [28] | 8-8-8 | 44.89/22.03/42.18 | 41.58/18.77/33.73 | 6-6-6 | 37.56/15.46/34.92 | 16.11/2.13/12.22 |
| Percentile [29] | 8-8-8 | 44.67/21.74/41.81 | 41.47/18.67/33.61 | 6-6-6 | 37.02/15.31/34.45 | 30.10/9.43/22.70 |
| EasyQuant [40] | 8-8-8 | 44.98/22.07/42.24 | 41.65/18.81/33.77 | 6-6-6 | 38.86/16.65/35.99 | 17.61/2.79/13.38 |
| **Ours** | 8-8-8 | **45.96/23.15/43.45** | **42.29/19.63/34.56** | 6-6-6 | **41.00/18.41/38.51** | **34.61/12.86/27.38** |

Table 7: PTQ results of BART model on summarization tasks in terms of ROUGE 1/2/L.

## 6    Conclusions and Discussions of Limitations

In this paper, we analyze the outlier phenomenon from the inducement and clipping impact on transformer language models. Based on these, we establish an outlier suppression framework to suppress the outliers. There also remain some open problems worthy of more in-depth investigations. For example, it is valuable to systematically explore whether the conclusion in this paper benefits other fields such as computer vision. Besides, as we supplement in the Appendix that the outliers occur not only in the fine-tuned (BERT) models but also in the pre-trained ones, it's also meaningful to dive into the pre-training process for a better understanding.

## Acknowledgment

We sincerely thank the anonymous reviewers for their serious reviews and valuable suggestions to make this better. This work was supported in part by National Natural Science Foundation of China under Grant 62022009 and Grant 61872021, and Beijing Nova Program of Science, and Technology under Grant Z191100001119050 and the Fundamental Research Funds for the Central Universities.

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
