# Appendix

Due to the space limitation of the main paper, we will provide supplementary analysis and experimental details in the appendix, including proof of equivalent transformation in Gamma Migration, illustration of quantization challenge, more analysis of outliers, supplementary experiments to better support our observations and methods, related works and implementation details.

## A  Supplementary illustration of Gamma Migration

In this section, we first put proof of the equivalent transformation Eq. (5). Then the detailed migration procedures of LayerNorm after the Feed Forward network (FFN) and Cross-Attention module are given. Especially, mark the LayerNorm after FFN as FFN-LN and the one after Multi-Head Attention as MHA-LN.

### A.1  Proof of equivalent transformation

To prove Eq. (5), we look at each element in the output of matrix multiplication. In detail, we mark the output as $\boldsymbol{h}$.

$$
\begin{aligned}
\boldsymbol{h}_i &= \sum_j \boldsymbol{W}_{i,j} \cdot (\boldsymbol{\gamma}_j \cdot \boldsymbol{x}_j) \\
&= \sum_j (\boldsymbol{\gamma}_j \cdot \boldsymbol{W}_{i,j}) \cdot \boldsymbol{x}_j.
\end{aligned}
\tag{10}
$$

Thus, for all the elements in $\boldsymbol{h}$, we have:

$$
\boldsymbol{W}(\boldsymbol{x} \odot \begin{bmatrix} \gamma_1 \\ \gamma_2 \\ ... \\ \gamma_n \end{bmatrix}) = (\boldsymbol{W} \odot \begin{bmatrix} \gamma_1 & \gamma_2 & ... & \gamma_n \\ \gamma_1 & \gamma_2 & ... & \gamma_n \\ ... \\ \gamma_1 & \gamma_2 & ... & \gamma_n \end{bmatrix})\boldsymbol{x},
\tag{11}
$$

The parameter $\boldsymbol{\gamma}$ is shared across samples and tokens, then the above equation always holds, and the weight in the next layer can absorb the $\boldsymbol{\gamma}$ naturally.

### A.2  Gamma Migration on other structures

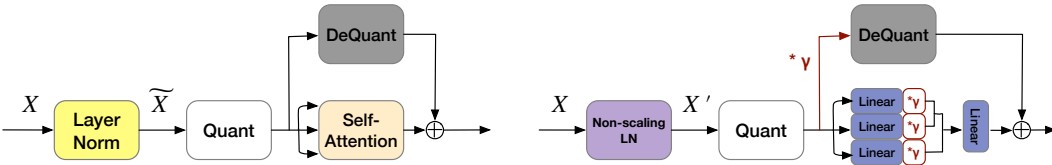

Figure 6: Comparison of the quantization flow before (left) and after (right) Gamma Migration in FFN-LN. The original LayerNorm = the Non-scaling LayerNorm * $\boldsymbol{\gamma}$.

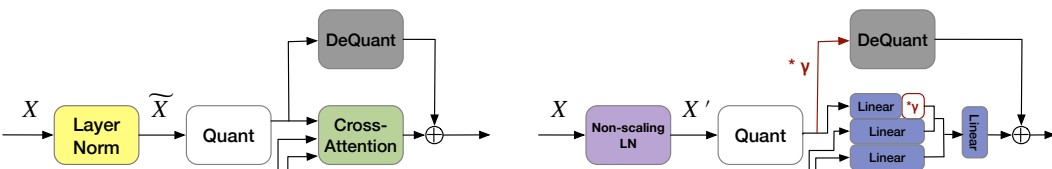

Figure 7: Comparison of the quantization flow before (left) and after (right) Gamma Migration in MHA-LN of the Cross-Attention module.

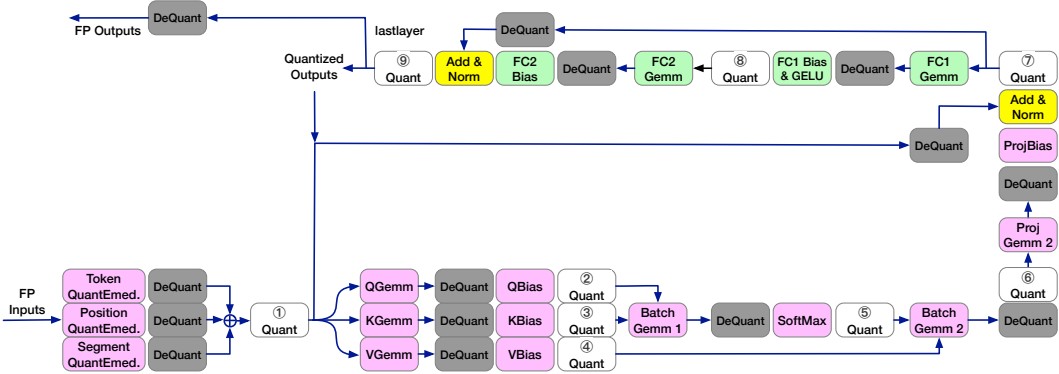

Figure 8: Position of activation quantization nodes. In real inference, the adjacent "DeQuant" and "Quant" operations will be merged into one "ReQuant" operation for faster computation.

# B  Quantization nodes

## B.1  Position of quantization nodes

For the position to insert quantization nodes, we find that different papers often have different choices, particularly at activation. This would bring difficulties for fair comparisons across methods and practical development on hardware.

By surveying multiple industry [38, 42] and academic solutions, we take the one in FasterTransformer [38]: Token (position, token type) embeddings are quantized to reduce the memory storage. Weights and activation engaged in matrix multiplication are also quantized. To be noted, we only give one quantizer to the same activation because it is friendly to hardware. Thus we will quantize the shortcut branch and take the same quantization parameter for the input of Query, Key, and Value modules, where some papers [24] do not and might suffer some problems on hardware.

A clear illustration about the position of activation quantization is depicted in Fig. 8. Here for ease of understanding, we mark each "Quant" node with a serial number and match them with the related module names in Table 8.

| ① | ② | ③ | ④ | ⑤ | ⑥ | ⑦ | ⑧ | ⑨ |
|---|---|---|---|---|---|---|---|---|
| Input Embedding | Query | Key | Value | Attention probs | Context | MHA-LN | GELU | FFN-LN |

Table 8: We map the label in Fig. 8 to the module name, which represents the quantization node inserted at the output of the corresponding module.

## B.2  Problematic quantization nodes

In this subsection, we give some simple and direct studies to elaborate on the most problematic tensors (outputs of LayerNorm structures and GELU). Verifications are done on fine-tuned BERT, RoBERTa, and encoder-decoder model BART.

On the one hand, we compare the cosine similarity between the FP value and the quantized one for each output. Activation nodes with cosine similarity lower than 0.99 are viewed as problematic positions (results in Table 9, Table 10). On another hand, we can observe the final accuracy recovery by disabling the quantization of each kind of activation. Both experiments indicate the obstacles when quantizing the outputs of LayerNorm and GELU.

| BERT-STS-B | | BERT-QQP | | BERT-MRPC | |
| --- | --- | --- | --- | --- | --- |
| output | cosine similarity (%) | output | cosine similarity (%) | output | cosine similarity (%) |
| layer.8.GELU | 87.83 | layer.9.GELU | 94.19 | layer.9.GELU | 92.00 |
| layer.11.GELU | 90.68 | layer.4.MHA-LN | 94.40 | layer.7.MHA-LN | 93.05 |
| layer.4.MHA-LN | 94.60 | layer.6.MHA-LN | 94.45 | layer.8.MHA-LN | 93.14 |
| layer.6.MHA-LN | 94.63 | layer.5.MHA-LN | 94.55 | layer.6.MHA-LN | 93.22 |
| layer.5.MHA-LN | 94.66 | layer.7.MHA-LN | 94.60 | layer.4.MHA-LN | 93.28 |
| layer.7.MHA-LN | 94.85 | layer.3.MHA-LN | 95.01 | layer.5.MHA-LN | 93.44 |
| layer.3.MHA-LN | 95.19 | layer.8.MHA-LN | 95.05 | layer.2.GELU | 93.94 |
| layer.10.MHA-LN | 95.45 | layer.2.GELU | 95.08 | layer.3.MHA-LN | 94.15 |
| layer.8.MHA-LN | 95.45 | layer.9.MHA-LN | 95.80 | layer.10.MHA-LN | 94.36 |
| layer.2.GELU | 95.48 | layer.10.MHA-LN | 96.13 | layer.9.MHA-LN | 94.58 |
| layer.9.MHA-LN | 95.60 | layer.1.MHA-LN | 96.84 | layer.8.GELU | 94.68 |
| layer.5.GELU | 96.86 | layer.0.MHA-LN | 96.87 | layer.10.GELU | 95.81 |
| layer.0.MHA-LN | 96.96 | layer.10.GELU | 97.02 | layer.0.MHA-LN | 96.99 |
| layer.1.MHA-LN | 97.15 | layer.2.MHA-LN | 97.50 | layer.1.MHA-LN | 97.12 |
| layer.9.GELU | 97.42 | layer.4.GELU | 97.57 | layer.2.MHA-LN | 97.66 |
| layer.4.GELU | 97.60 | layer.5.GELU | 97.71 | layer.11.GELU | 97.70 |
| layer.2.MHA-LN | 97.67 | layer.3.GELU | 98.30 | layer.5.GELU | 97.91 |
| layer.6.GELU | 98.07 | layer.11.GELU | 98.43 | layer.4.GELU | 98.04 |
| layer.3.GELU | 98.22 | layer.1.GELU | 98.46 | layer.11.MHA-LN | 98.16 |
| layer.1.GELU | 98.34 | layer.0.GELU | 98.60 | layer.1.GELU | 98.18 |
| layer.7.GELU | 98.43 | layer.8.GELU | 98.63 | layer.0.GELU | 98.31 |
| layer.10.GELU | 98.44 | layer.7.GELU | 98.69 | layer.7.GELU | 98.42 |
| layer.0.GELU | 98.52 | layer.11.MHA-LN | 98.76 | layer.3.GELU | 98.67 |
| layer.11.MHA-LN | 98.60 | layer.6.GELU | 98.77 | layer.6.GELU | 98.74 |
| layer.9.FFN-LN | 98.79 | layer.10.Context | 98.96 | layer.10.FFN-LN | 98.94 |

| RoBERTa-MNLI | | RoBERTa-QNLI | | RoBERTa-QQP | |
| --- | --- | --- | --- | --- | --- |
| output | cosine similarity(%) | output | cosine similarity(%) | output | cosine similarity(%) |
| layer.7.GELU | 93.91 | layer.10.GELU | 90.08 | layer.2.GELU | 93.56 |
| layer.9.GELU | 94.25 | layer.7.GELU | 91.60 | layer.3.GELU | 94.27 |
| layer.2.GELU | 94.64 | layer.5.GELU | 95.58 | layer.4.GELU | 95.96 |
| layer.10.GELU | 94.79 | layer.4.GELU | 95.59 | layer.1.GELU | 96.69 |
| layer.8.GELU | 94.83 | layer.2.GELU | 95.89 | layer.5.GELU | 96.71 |
| layer.5.GELU | 96.16 | layer.8.GELU | 96.02 | layer.0.GELU | 97.04 |
| layer.4.GELU | 96.28 | layer.3.GELU | 96.33 | layer.0.MHA-LN | 97.09 |
| layer.1.GELU | 96.38 | layer.1.GELU | 96.52 | layer.7.GELU | 97.41 |
| layer.3.GELU | 96.69 | layer.9.GELU | 96.85 | layer.1.MHA-LN | 97.59 |
| layer.6.GELU | 96.82 | layer.11.MHA-LN | 97.00 | layer.8.GELU | 97.81 |
| layer.0.MHA-LN | 97.16 | layer.0.MHA-LN | 97.13 | layer.8.FFN-LN | 98.10 |
| layer.11.MHA-LN | 97.26 | layer.6.GELU | 97.36 | layer.7.FFN-LN | 98.13 |
| layer.0.GELU | 97.30 | layer.0.GELU | 97.49 | layer.0.FFN-LN | 98.16 |
| layer.10.FFN-LN | 97.64 | layer.1.MHA-LN | 97.66 | layer.1.FFN-LN | 98.23 |
| layer.10.MHA-LN | 97.64 | layer.8.Context | 97.67 | layer.6.FFN-LN | 98.28 |
| layer.1.MHA-LN | 97.67 | layer.10.FFN-LN | 97.72 | layer.6.GELU | 98.29 |
| layer.9.FFN-LN | 97.84 | layer.10.MHA-LN | 97.75 | layer.7.MHA-LN | 98.32 |
| layer.8.FFN-LN | 97.90 | layer.9.Context | 97.79 | layer.8.MHA-LN | 98.33 |
| layer.7.FFN-LN | 98.05 | layer.9.FFN-LN | 97.89 | layer.6.MHA-LN | 98.35 |
| layer.9.MHA-LN | 98.11 | layer.8.FFN-LN | 97.92 | layer.5.FFN-LN | 98.36 |
| layer.8.MHA-LN | 98.13 | layer.7.FFN-LN | 97.99 | layer.2.MHA-LN | 98.42 |
| layer.0.FFN-LN | 98.14 | layer.0.FFN-LN | 98.14 | layer.5.MHA-LN | 98.43 |
| layer.6.FFN-LN | 98.25 | layer.8.MHA-LN | 98.15 | layer.4.FFN-LN | 98.46 |
| layer.1.FFN-LN | 98.33 | layer.9.MHA-LN | 98.17 | layer.3.MHA-LN | 98.49 |
| layer.5.FFN-LN | 98.34 | layer.6.FFN-LN | 98.19 | layer.4.MHA-LN | 98.50 |
| layer.6.MHA-LN | 98.36 | layer.5.FFN-LN | 98.26 | layer.2.FFN-LN | 98.50 |
| layer.7.MHA-LN | 98.36 | layer.6.MHA-LN | 98.28 | layer.9.FFN-LN | 98.52 |
| layer.4.FFN-LN | 98.39 | layer.7.MHA-LN | 98.31 | layer.3.FFN-LN | 98.57 |
| layer.5.MHA-LN | 98.43 | layer.1.FFN-LN | 98.32 | layer.10.GELU | 98.58 |
| layer.4.MHA-LN | 98.47 | layer.4.FFN-LN | 98.34 | layer.9.MHA-LN | 98.60 |
| layer.3.FFN-LN | 98.48 | layer.5.MHA-LN | 98.37 | layer.10.FFN-LN | 98.60 |
| layer.2.MHA-LN | 98.50 | layer.3.FFN-LN | 98.45 | layer.11.MHA-LN | 98.75 |
| layer.2.FFN-LN | 98.54 | layer.4.MHA-LN | 98.45 | layer.9.GELU | 98.86 |
| layer.3.MHA-LN | 98.55 | layer.2.FFN-LN | 98.50 | layer.10.MHA-LN | 98.89 |
| | | layer.2.MHA-LN | 98.52 | | |
| | | layer.3.MHA-LN | 98.55 | | |

Table 9: The sorted cosine similarity between the output and the quantized one (6-bit) on BERT and RoBERTa models. We aim at the most problematic ones with cosine similarity lower than 99%.

| BART-CNN/DailyMail | | BART-XSum | |
|---|---|---|---|
| output | cosine similarity (%) | output | cosine similarity (%) |
| layers.4.GELU (Decoder) | 67.96 | layers.3.GELU (Decoder) | 74.37 |
| layers.3.GELU (Decoder) | 69.50 | layers.4.GELU (Decoder) | 75.05 |
| layers.4.MHA-LN (Encoder-Decoder) | 76.03 | layers.2.GELU (Decoder) | 82.36 |
| layers.2.GELU (Decoder) | 76.05 | layers.4.MHA-LN (Encoder-Decoder) | 82.84 |
| layers.2.MHA-LN (Encoder-Decoder) | 77.88 | layers.1.MHA-LN (Encoder) | 83.04 |
| layers.0.GELU (Decoder) | 80.83 | layers.2.MHA-LN (Encoder-Decoder) | 84.31 |
| layers.5.MHA-LN (Encoder) | 84.20 | layers.4.MHA-LN (Encoder) | 84.53 |
| layers.1.MHA-LN (Encoder) | 84.33 | layers.5.MHA-LN (Encoder) | 84.69 |
| layers.1.MHA-LN (Encoder-Decoder) | 85.01 | layers.3.MHA-LN (Encoder) | 86.47 |
| layers.4.MHA-LN (Encoder) | 85.03 | layers.1.MHA-LN (Encoder-Decoder) | 86.97 |
| layers.3.MHA-LN (Encoder-Decoder) | 86.78 | layers.0.MHA-LN (Encoder) | 87.69 |
| layers.3.MHA-LN (Encoder) | 87.12 | layers.0.GELU (Decoder) | 87.77 |
| layers.0.MHA-LN (Encoder) | 87.30 | layers.3.MHA-LN (Encoder-Decoder) | 88.11 |
| layers.1.GELU (Decoder) | 87.61 | layers.2.MHA-LN (Encoder) | 89.14 |
| layers.2.MHA-LN (Encoder) | 89.64 | layers.0.GELU (Encoder) | 92.21 |
| layers.5.GELU (Decoder) | 91.78 | layers.1.GELU (Decoder) | 93.60 |
| layers.0.MHA-LN (Encoder-Decoder) | 93.62 | layers.0.MHA-LN (Encoder-Decoder) | 93.61 |
| layers.0.GELU (Encoder) | 95.09 | layers.5.FFN-LN (Decoder) | 95.44 |
| layers.2.GELU (Encoder) | 95.91 | layers.5.GELU (Decoder) | 96.35 |
| layers.3.GELU (Encoder) | 96.44 | layers.3.GELU (Encoder) | 96.41 |
| layers.3.MHA-LN (Decoder) | 96.90 | layers.2.GELU (Encoder) | 96.57 |
| layers.5.MHA-LN (Decoder) | 97.46 | layers.3.MHA-LN (Decoder) | 96.87 |
| layers.2.Context (Encoder-Decoder) | 97.51 | layers.2.Context (Encoder-Decoder) | 96.99 |
| layers.5.MHA-LN (Encoder-Decoder) | 97.71 | layers.1.MHA-LN (Encoder) | 97.20 |
| layers.4.FFN-LN (Decoder) | 97.83 | layers.5.MHA-LN (Encoder-Decoder) | 97.56 |
| layers.4.GELU (Encoder) | 97.85 | layers.0.Context (Encoder-Decoder) | 97.72 |
| layers.1.GELU (Encoder) | 97.88 | layers.5.GELU (Encoder) | 97.74 |
| layers.5.GELU (Encoder) | 97.97 | layers.4.FFN-LN (Decoder) | 98.02 |
| layers.5.FFN-LN (Decoder) | 98.32 | layers.4.GELU (Encoder) | 98.04 |
| layers.2.Context (Decoder) | 98.40 | layers.0.Context (Decoder) | 98.11 |
| layers.1.MHA-LN (Decoder) | 98.51 | layers.5.MHA-LN (Decoder) | 98.20 |
| layers.3.FFN-LN (Decoder) | 98.52 | layers.2.FFN-LN (Decoder) | 98.28 |
| layers.0.Context (Decoder) | 98.53 | layers.3.Context (Encoder-Decoder) | 98.31 |
| layers.4.MHA-LN (Decoder) | 98.54 | layers.1.Context (Decoder) | 98.32 |
| layers.2.MHA-LN (Decoder) | 98.63 | layers.3.FFN-LN (Decoder) | 98.36 |
| layers.2.FFN-LN (Decoder) | 98.66 | layers.1.MHA-LN (Decoder) | 98.38 |
| layers.1.FFN-LN (Decoder) | 98.71 | layers.5.Context (Encoder-Decoder) | 98.46 |
| layers.0.Context (Encoder-Decoder) | 98.71 | layers.2.Context (Decoder) | 98.56 |
| layers.0.FFN-LN (Decoder) | 98.72 | layers.4.MHA-LN (Decoder) | 98.58 |
| layers.5.Context (Encoder-Decoder) | 98.72 | layers.2.MHA-LN (Decoder) | 98.64 |
| layers.4.Context (Decoder) | 98.92 | layers.0.FFN-LN (Decoder) | 98.71 |
| layers.0.MHA-LN (Decoder) | 98.93 | layers.1.FFN-LN (Decoder) | 98.72 |
| | | layers.0.MHA-LN (Decoder) | 98.80 |

Table 10: The sorted cosine similarity between the output and the quantized one (6-bit) on BART models. We aim at the most problematic ones with cosine similarity lower than 99%.

| Model | 32-32-32 | 6-6-6 | Input Embedding | Query | Key | Value | Attention probs | Context | MHA-LN | GELU | FFN-LN |
|---|---|---|---|---|---|---|---|---|---|---|---|
| BERT-MRPC | 87.75 | 31.86 | 31.62 | 32.11 | 32.11 | 32.6 | 31.62 | 31.86 | **83.09** | **34.31** | 31.86 |
| BERT-QQP | 90.91 | 69.0 | 69.22 | 69.05 | 68.95 | 69.24 | 68.09 | 69.25 | **88.93** | **74.25** | 70.01 |
| BERT-STS-B | 89.70 | 59.79 | 57.8 | 57.61 | 58.2 | 56.45 | 54.02 | 55.12 | **84.1** | **79.6** | 53.68 |
| RoBERTa-MNLI | 87.75 | 34.90 | 36.05 | 35.69 | 35.54 | 35.27 | 35.68 | 36.08 | **66.93** | **60.50** | **53.82** |
| RoBERTa-QNLI | 92.68 | 62.13 | 65.04 | 65.77 | 64.23 | 64.73 | 64.54 | 64.42 | **84.55** | **69.71** | **76.66** |
| RoBERTa-QQP | 91.6 | 74.37 | 76.24 | 75.97 | 76.01 | 76.41 | 75.50 | 75.92 | **87.80** | **84.28** | **80.89** |

Table 11: Influence study of quantization nodes. The comparisons of the second and third columns show the performance drop with 6-bit MinMax calibration and quantization. The subsequent columns show the recovered performance after disabling the quantization of a certain kind of output defined in Table 8, which implies the effect of quantizing this node. For example, "Query" means disabling the quantization of output at Query modules across 12 layers. Obvious improvements are marked in bold.

## C  Analysis of outliers

### C.1  Outlier phenomenon

By going deeper into the above problematic activations, we find that large outliers in them cause the large quantization error, and these outliers present some structured features from the embedding

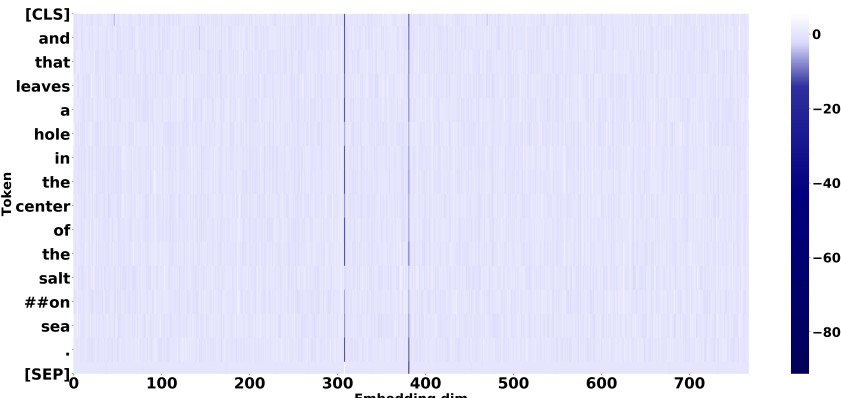

Figure 9: The dark strips on embedding dim 308 and 381 represent the outliers across almost all tokens at LayerNorm's output in BERT-SST-2.

and token perspectives. Activations of almost all tokens attend to outliers in specific embedding dimensions like 308 and 381 embedding dimensions in Fig. 9. Upon these dimensions, some tokens like the [SEP] token in Fig. 10 attend to even more aggressive outliers compared to other tokens in (Fig. 10). In fact, we find this often happens on token [SEP], [CLS], punctuations like commas and periods, and other high-frequency tokens like "the", "and", "of".

## C.2    Detailed discussion about the inducement

Here, we discuss the inducement of the outlier phenomenon from embedding and token perspectives.

For the embedding phenomenon, the Sec. 3.1 has explained the scaling parameter amplifies the outliers at certain embeddings. In fact, we find that this not only emerges in fine-tuned models but is also obvious in the pre-trained ones. By injecting constraints such as weight decay or kurtosis regularization [43] to LayerNorm's parameter when fine-tuning the FP model, it is still hard to suppress the aggressive values in the scaling parameter without affecting FP performance. Hence, we conjecture that this phenomenon is beneficial to the FP performance though it indeed brings challenges to quantization.

Moreover, the huge deviation in the token range we think is caused by the token frequency in the pre-training phase. Because we find the tokens which hold more aggressive signals occur frequently during pre-training like [SEP], [CLS] occur in each example, and '.' is often used in an expression. We also notice that these tokens' word (token) embeddings have larger values than others. According to these, a possible explanation might be like: the frequency information biases the word embedding

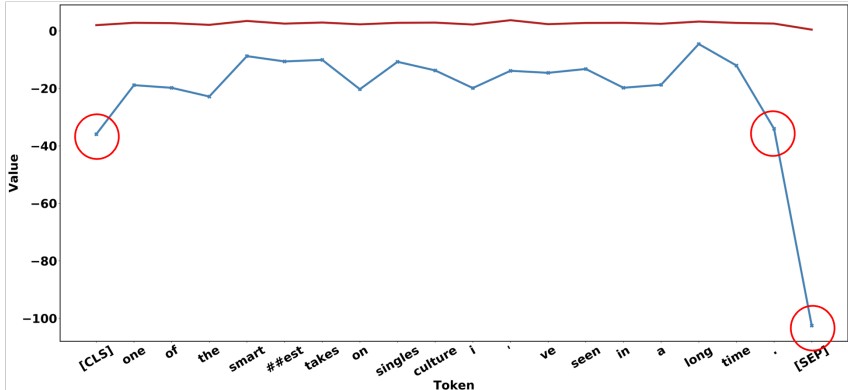

Figure 10: We draw the token range and token [SEP], [CLS], '.' attend to sharper outliers here as marked in red circles.

space and brings different features. The sharper outliers spread to subsequent layers and seem to be less important as indicated in Sec. 3.2. Therefore, we conjecture that a good word embedding without being biased by frequency information can behave better in quantization. But we can find those less important outliers in an efficient way and clip them as well. This suits better for post-training quantization without large-scale re-training.

For the inducement of outliers, note that [44] also mentioned the connection between the scaling parameter and outliers in the last LayerNorm each BERT layer. But we emphasize the amplification effect of the scaling parameter, especially for the LayerNorm after Multi-Head Attention. This naturally generates the finding of quantization-friendly distribution contributed by removing the scaling parameter. About the unbalanced token frequency, a concurrent work [45] explores carefully from the FP performance perspective.

# D   Supplementary experiments

## D.1   Supplementary evidence of outliers in LayerNorm

We show more evidence of the same outlier phenomenon in LayerNorm and illustrate that the output of Non-scaling LayerNorm is more quantization-friendly than the normal one. Firstly, Fig. 11 and Fig. 12 are presented to build a formal understanding, where the $X'$ has weaker outliers. Furthermore, more quantitative results about cosine similarity are put in Table 12 to indicate the improvement on the most problematic tensors Sec. B.2 brought by extracting the scaling parameter $\gamma$. Here, we discuss the inducement of the outlier phenomenon from embedding and token perspectives.

## D.2   Supplementary evidence of clipping impact

We provide more evidence of accuracy and token impact by clipping the outputs to different levels in Table 13.

The first thing is that different outliers have very different importance, where some very large values can be clipped sharply but will not introduce large accuracy degradation, whereas the performance decreases quickly with some being clipped. For example, for the outputs of MHA-LN, clipping them from -60 to -45 seems reliable in the FP model and of course friendly in the quantized one. However, clipping from -40 to -35 will induce about 5% performance loss.

Another key point is that those large outliers only belong to several tokens regarding the big divergence of the token range. For example, for values in (-60, -45), the clipped tokens are still 3% for most of the layers. Thus, finding the clipping range from the token perspective can help to jump over the less important area quickly.

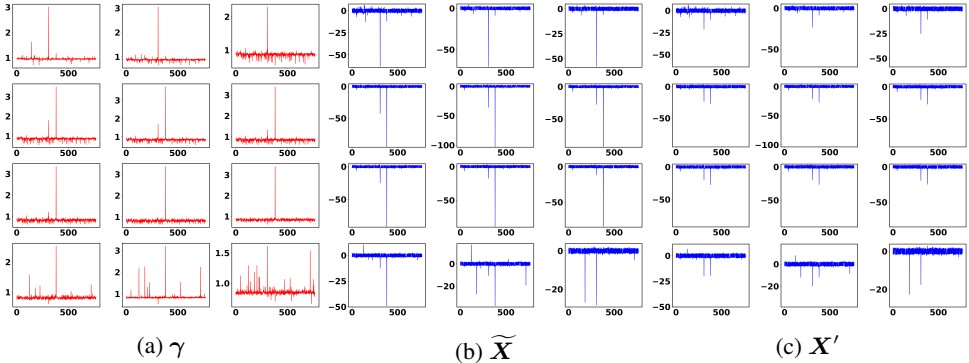

(a) $\gamma$          (b) $\widetilde{X}$          (c) $X'$

Figure 11: $\gamma$, $\widetilde{X}$ and $X'$ across 12 MHA-LN in BERT-SST-2, where $\widetilde{X} = \gamma \odot X'$. For the latter two, we draw the highest-magnitude value at each embedding dim. It can be seen that $X'$ holds milder distribution.

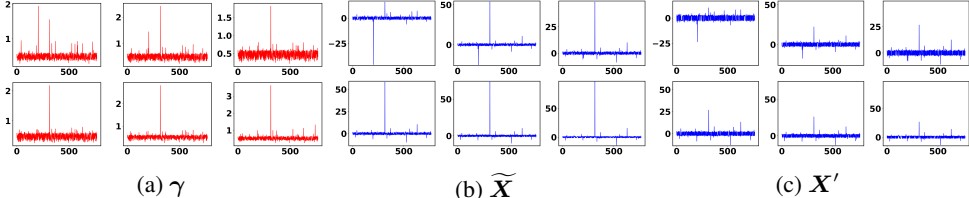

(a) $\gamma$      (b) $\widetilde{X}$      (c) $X'$

Figure 12: $\gamma$, $\widetilde{X}$ and $X'$ across 6 LayerNorm in BART-QQP, where $\widetilde{X} = \gamma \odot X'$. For the latter two, we draw the highest-magnitude value at each embedding dim.

| Model | 0 | 1 | 2 | 3 | 4 | 5 | 6 | 7 | 8 | 9 | 10 | 11 |
|---|---|---|---|---|---|---|---|---|---|---|---|---|
| **BERT-MRPC** | | | | | | | | | | | | |
| MHA-LN | +2.24 | +2.17 | +1.48 | +4.84 | +5.70 | +5.55 | +5.76 | +5.83 | +5.56 | +4.32 | +4.79 | +0.85 |
| **BERT-QQP** | | | | | | | | | | | | |
| MHA-LN | +2.35 | +2.35 | +1.61 | +4.00 | +4.58 | +4.43 | +4.51 | +4.24 | +3.64 | + 3.07 | +3.19 | +0.24 |
| **BERT-STS-B** | | | | | | | | | | | | |
| MHA-LN | +2.19 | +2.03 | +1.43 | +3.82 | +4.39 | +4.34 | +4.35 | +4.03 | +3.25 | +3.33 | +3.44 | +0.51 |
| **RoBERTa-MNLI** | | | | | | | | | | | | |
| MHA-LN | +1.49 | +0.81 | +0.25 | +0.18 | +0.16 | + 0.16 | +0.22 | +0.19 | +0.25 | + 0.31 | +0.59 | +1.17 |
| FFN-LN | +0.31 | + 0.43 | +0.16 | +0.24 | +0.25 | +0.27 | +0.28 | +0.31 | +0.34 | +0.43 | +0.49 | +0.04 |
| **RoBERTa-QNLI** | | | | | | | | | | | | |
| MHA-LN | +1.62 | +0.88 | +0.25 | +0.19 | +0.17 | +0.18 | +0.22 | +0.18 | +0.23 | +0.24 | +0.52 | +1.31 |
| FFN-LN | +0.33 | +0.47 | +0.22 | +0.25 | +0.26 | +0.30 | +0.28 | +0.32 | +0.31 | +0.36 | +0.49 | +0.53 |
| **RoBERTa-QQP** | | | | | | | | | | | | |
| MHA-LN | +1.57 | +0.93 | +0.32 | +0.25 | +0.21 | +0.22 | +0.29 | +0.33 | +0.43 | + 0.39 | +0.30 | +0.64 |
| FFN-LN | +0.32 | +0.52 | +0.16 | +0.24 | +0.27 | +0.33 | +0.33 | +0.42 | +0.49 | +0.33 | +0.45 | +0.20 |
| **BART-CNN/DailyMail** | | | | | | | | | | | | |
| MHA-LN (Encoder) | +11.26 | +14.07 | +8.81 | +11.25 | +13.86 | +14.13 | | | | | | |
| MHA-LN (Decoder) | +0.23 | +0.19 | +0.01 | +1.69 | +0.23 | +1.29 | | | | | | |
| MHA-LN (Encoder-Decoder) | +5.21 | +13.82 | +20.94 | +11.94 | +22.74 | +1.04 | | | | | | |
| FFN-LN (Decoder) | +0.20 | +0.14 | +0.03 | +0.17 | +0.04 | +0.21 | | | | | | |
| **BART-XSum** | | | | | | | | | | | | |
| MHA-LN (Encoder) | +10.90 | +15.07 | +9.17 | +11.77 | +13.81 | +13.58 | | | | | | |
| MHA-LN (Decoder) | +0.15 | +0.12 | +0.09 | +1.63 | +0.23 | +0.61 | | | | | | |
| MHA-LN (Encoder-Decoder) | +5.09 | +11.75 | +14.50 | +10.57 | +15.96 | +1.32 | | | | | | |
| FFN-LN (Decoder) | +0.21 | +0.04 | +0.34 | +0.23 | +0.54 | +0.38 | | | | | | |

Table 12: Cosine similarity (%) improvement after extracting $\gamma$ in LayerNorm. This metric is evaluated on 256 samples from dev set. (BART only has 6 layers and thus the right half is left empty. )

### D.3 Comparisons among Token-Wise Clipping and existing methods

We compare the coarse stage of Token-Wise Clipping with OMSE, percentile, and direct step size learning and argue that ours is more effective Table 14 and efficient Table 15.

Our Token-Wise Clipping searches superior clipping ratio towards the final performance and works in a remarkably efficient way (Reasons have been explained in Sec. 4.2) with about 2 minutes evaluating 30 ratios on GLUE tasks.

On the contrary, OMSE only minimizes the local quantization error and behaves terribly. For instance, it calculates 40 as the best clipping range for the distribution presented in Fig. 2 while 10 is much better. Also, OMSE runs very slowly even with the fast golden section search.

For the direct step size learning and Percentile methods, though they consider the final loss for the clipping range, they still suffer some problems in the case that the unimportant outliers can cover a large area. Direct step size learning without a good initialization point needs a proper learning rate and much tuning time to achieve the key part. Take an extreme case as an example. In QAT, step size has been tuned sufficiently but we still notice that the quantized model can be further clipped. Besides, as the Percentile builds a histogram of the activation and searches for the best clipping ratio from the value perspective, it is time-costly to jump over the relatively unimportant outliers.

### D.4 Supplementary results of QAT

We apply our methods to RoBERTa and BART on quantization-aware training. From Table 16, on RoBERTa, ours still surpasses LSQ+ by 2.54% on QNLI, 7.53% on STS-B. On BART models,

| Clipping Value | Accuracy | 0 | 1 | 2 | 3 | 4 | 5 | 6 | 7 | 8 | 9 | 10 | 11 |
|---|---|---|---|---|---|---|---|---|---|---|---|---|---|
| BERT-MRPC (GELU) | 87.75 | | | | | | | | | | | | |
| 80.0 | 87.25 | 0.00 | 0.00 | 0.00 | 0.00 | 0.00 | 0.00 | 0.00 | 0.00 | 0.00 | 0.15 | 4.33 | 0.00 |
| 60.0 | 87.25 | 0.00 | 0.00 | 1.88 | 0.00 | 0.00 | 0.00 | 0.00 | 0.00 | 0.00 | 0.31 | 4.58 | 0.00 |
| 40.0 | 87.01 | 0.00 | 0.00 | 3.76 | 0.00 | 0.00 | 0.00 | 0.00 | 0.00 | 0.00 | 2.29 | 4.68 | 0.00 |
| 20.0 | 87.01 | 0.00 | 0.00 | 3.76 | 0.00 | 0.00 | 0.00 | 0.00 | 0.00 | 0.01 | 3.65 | 4.79 | 0.00 |
| 10.0 | 87.25 | 4.64 | 1.36 | 3.76 | 0.00 | 1.73 | 3.77 | 0.00 | 0.00 | 0.15 | 4.53 | 4.84 | 0.04 |
| 5.0 | 87.25 | 49.88 | 19.6 | 7.11 | 7.89 | 13.99 | 26.68 | 34.13 | 19.79 | 15.1 | 5.00 | 4.98 | 45.9 |
| 2.0 | **84.07** | 99.17 | 98.87 | 98.81 | 98.9 | 97.62 | 97.38 | 97.01 | 96.39 | 94.54 | 75.55 | 45.77 | 92.98 |
| 1.5 | 78.92 | 99.96 | 99.98 | 99.94 | 99.94 | 99.77 | 99.83 | 99.77 | 99.76 | 99.75 | 94.72 | 78.11 | 98.01 |
| BERT-QNLI (MHA-LN) | 91.84 | | | | | | | | | | | | |
| -60 | 91.67 | 1.96 | 2.68 | 0.00 | 3.92 | 3.92 | 3.92 | 3.92 | 3.92 | 3.92 | 0.00 | 0.00 | 0.00 |
| -55 | 91.69 | 1.96 | 6.21 | 1.96 | 3.92 | 3.92 | 3.92 | 3.92 | 3.92 | 3.92 | 0.22 | 0.00 | 0.00 |
| -50 | 91.43 | 2.91 | 10.69 | 1.97 | 3.92 | 3.92 | 3.92 | 3.92 | 3.92 | 3.92 | 3.88 | 0.06 | 0.00 |
| -45 | 91.25 | 9.85 | 16.15 | 7.17 | 3.92 | 3.92 | 3.92 | 3.92 | 3.92 | 3.92 | 3.92 | 0.45 | 0.00 |
| -40 | 90.28 | 16.96 | 23.13 | 13.61 | 5.71 | 3.92 | 3.92 | 3.92 | 3.92 | 4.32 | 4.23 | 1.46 | 0.00 |
| -35 | **85.54** | 22.51 | 29.36 | 23.42 | 7.46 | 4.50 | 3.92 | 3.92 | 3.92 | 5.52 | 5.57 | 3.21 | 0.01 |
| -30 | 78.36 | 27.48 | 36.67 | 32.87 | 9.39 | 7.92 | 3.92 | 3.92 | 4.04 | 6.73 | 5.86 | 6.46 | 8.19 |
| -25 | 72.73 | 34.81 | 42.43 | 41.88 | 23.05 | 15.63 | 6.55 | 3.92 | 6.03 | 8.16 | 5.99 | 8.47 | 13.84 |
| -20 | 72.52 | 41.74 | 47.64 | 49.76 | 37.66 | 33.98 | 13.51 | 6.55 | 8.31 | 19.08 | 6.94 | 10.29 | 40.53 |

| Clipping Value | Accuracy | 0 | 1 | 2 | 3 | 4 | 5 | 0 | 1 | 2 | 3 | 4 | 5 |
|---|---|---|---|---|---|---|---|---|---|---|---|---|---|
| BART-CoLA (GELU) | 56.32 | | | | | | | | | | | | |
| 80.0 | 56.32 | 0.00 | 0.00 | 0.00 | 0.00 | 0.00 | 0.00 | 8.48 | 8.60 | 0.00 | 0.00 | 0.00 | 0.00 |
| 60.0 | 56.32 | 0.00 | 0.00 | 0.00 | 0.00 | 0.00 | 0.00 | 8.60 | 8.60 | 0.00 | 0.00 | 0.00 | 0.00 |
| 40.0 | 56.32 | 0.00 | 0.00 | 0.00 | 0.00 | 0.00 | 0.00 | 8.60 | 8.60 | 8.60 | 0.00 | 8.60 | 0.00 |
| 20.0 | 56.32 | 0.01 | 0.00 | 0.00 | 0.00 | 0.00 | 0.00 | 17.21 | 17.21 | 8.60 | 8.61 | 8.61 | 0.00 |
| 10.0 | 56.32 | 8.60 | 4.34 | 8.60 | 8.60 | 0.00 | 0.00 | 17.21 | 17.21 | 8.60 | 8.61 | 8.61 | 8.60 |
| 5.0 | 56.58 | 9.31 | 8.80 | 8.83 | 8.79 | 0.13 | 0.43 | 20.3 | 17.23 | 8.74 | 8.80 | 8.87 | 9.29 |
| 2.0 | **54.06** | 92.49 | 90.98 | 78.52 | 70.7 | 79.58 | 62.35 | 97.27 | 92.14 | 74.54 | 59.41 | 53.88 | 42.17 |
| 1.5 | 52.37 | 98.98 | 98.59 | 96.46 | 94.45 | 96.63 | 86.5 | 99.88 | 99.38 | 95.1 | 87.58 | 84.8 | 72.38 |

Table 13: We evaluate the accuracy directly on dev set with output activation cut by the clipping value. The subsequent columns records the ratio of clipped tokens to all tokens each layer. For BART, we also consider the GELU module in Decoder. Bold numbers show the inflection point of accuracy change.

| Method | CoLA (Matt.) | MNLI (acc m/mm) | MRPC (f1/acc) | QNLI (acc) | QQP (f1/acc) | RTE (acc) | SST-2 (acc) | STS-B (Pear./Spear.) |
|---|---|---|---|---|---|---|---|---|
| RoBERTa (FP) | 62.50 | 87.75/87.23 | 93.1/90.44 | 92.68 | 88.78/91.6 | 80.51 | 95.18 | 91.04/90.72 |
| OMSE [28] | 1.81 | 72.89/72.65 | 85.38/78.68 | 76.53 | 85.24/88.94 | 64.26 | 91.17 | 80.81/81.99 |
| Step size learning [12] | 4.64 | 71.77/73.18 | 85.42/79.17 | 77.28 | 85.19/88.91 | 65.34 | 90.71 | 80.23/81.25 |
| Percentile [46] | 20.73 | 72.23/73.68 | 84.83/78.43 | 77.16 | 82.21/87.44 | 62.82 | 88.19 | 79.41/79.64 |
| Token-Wise Clipping (Coarse Stage) | **34.95** | **80.56/80.84** | **85.05/79.41** | **79.46** | **85.96/89.31** | **66.43** | **91.63** | **82.03/82.45** |
| BERT (FP) | 59.60 | 84.94/84.76 | 91.35/87.75 | 91.84 | 87.82/90.91 | 72.56 | 93.35 | 89.70/89.28 |
| OMSE | 35.44 | 74.00/73.30 | 81.54/76.47 | 84.66 | 76.07/82.12 | 64.26 | 86.27 | 85.57/86.05 |
| Step size learning | 35.77 | 74.11/73.76 | 82.95/77.94 | 85.19 | 75.79/81.91 | 64.62 | 87.16 | 85.78/86.47 |
| Percentile | 37.32 | 72.40/71.69 | 85.09/79.90 | 79.37 | 72.58/80.19 | 61.73 | 87.27 | 86.38/87.29 |
| Token-Wise Clipping (Coarse Stage) | **47.21** | **77.53/78.01** | **85.40/80.39** | **86.47** | **74.98/83.88** | **64.62** | **91.17** | **86.48/87.06** |

Table 14: Comparisons among existing techniques and the coarse stage of Token-Wise Clipping on 6-bit BERT and RoBERTa models. For the percentile, we search its hyper-parameter in [0.999, 0.9999, 0.99999] and report the best one on dev set. It can be seen that only the coarse stage of our method has surpasses others.

we achieve an absolute improvement of 1.73–32.11 points against the best baseline. The outlier suppression framework can be extended to other applications, such as integer-only quantization [25] as well, which proposes the polynomial approximation of non-linear operations for Transformer-based models.

# E  Supplementary related works

Quantization algorithms are usually grouped into two categories: (1) Quantization-Aware Training (QAT) and (2) Post-Training Quantization (PTQ). The former fine-tunes the FP model to low-bit and embraces good outcomes with awareness of quantization during training. Apart from learning weight for better performance, [41, 11] propose to learn the quantization parameters. The latter, PTQ, usually conducts fast calibration on the FP model with much less computation and fewer data. [28] transforms quantization to a Minimum Mean Squared Error problem. [40] alternately optimizes the step size of weight and activation towards the matrix multiplication output.

| | OMSE | | Percentile | Token-Wise Clipping (Coarse Stage) |
|---|---|---|---|---|
| Grid Search (30 iterations) | Golden Section Search | search (3 times) | | Grid Search (30 iterations) |
| 1754s | 439.29s | 301.49s | | 135.73s |

Table 15: The time of activation calibration on 256 samples of each algorithm. As direct step size learning takes OMSE as its initialization, we do not compare the time here.

Recently, quantization has become popular in Transformer-based models. For quantization-aware training, [21] explores 8-bit quantization on BERT-like models. [22] adopts group-wise quantization and applies mixed-precision quantization based on the Hessian information. [23] investigates various distillation losses on BERT and combines the distillation with quantization. [25] approximates the nonlinear function in Transformer architectures to enjoy integer-only inference. [47] quantizes a different random subset of weights each forward pass during training to decrease quantization noise. Moreover, [48] explores underlying difficulties of quantizing generative models. Due to the sequential computation nature of this type of model, they find that word embedding is easier to be homogeneous and devise a token-level contrastive distillation method to combat this obstacle. For post-training quantization, [26] notices the structured outliers in Transformer-based models with the occurrence at a few embedding dims and the special separator token. They point out that the high dynamic ranges will even hurt the 8-bit quantization performance and suggest taking per-embedding-group quantization for this unique challenge. While they walk around the problem and their method brings extra computation burden, we explore the inducement and clipping impact of these structured outliers and solve them without computation overhead.

# F    Supplementary implementation details

For quantizer details, we insert quantization nodes as Sec. B.1. We adopt symmetric per-channel quantization on weight and asymmetric per-layer quantization on activation.

For PTQ experiments, we sample 256 examples as the calibration dataset with batch size set to 32 on GLUE benchmark and SQuAD, 4 for CNN/DailyMail and XSum. For learning in the fine-grained stage of the Token-Wise Clipping, we always tune 3 epochs with learning rate 1e-5 across datasets because the first step already produces good outcomes.

For QAT experiments on the GLUE benchmark, we equip our methods with LSQ+ [12]. The coarse-grained stage of Token-Wise Clipping is used to initialize quantization parameters, the fine-grained stage is removed because LSQ+ has armed with step size learning. About hyper-parameters, learning rate is searched in {1e-5, 2e-5, 3e-5, 4e-5, 5e-5}. Batch size is usually set to 32 unless smaller (8 and 16) ones are also tried on small datasets including CoLA, MRPC, RTE, and STS-B. As for epochs, we follow [26] on BERT (3 epochs for MNLI and QQP, 6 epochs for others), [25] on RoBERTa (6 epochs for MNLI and QQP, 12 epochs for others), and take 6 or 12 epochs on BART as well. Other hyper-parameters are inspected and kept fixed across datasets including self-attention dropout rate 0.1, hidden states dropout rate 0.0, weight decay 0.0, and warmup ratio 10%. For baseline mechanisms like LSQ+ and PACT, we conduct the above learning rate and batch size search as well for fair comparisons.

| Method | Bits (W-E-A) | CoLA (Matt.) | MNLI (acc m/mm) | MRPC (f1/acc) | QNLI (acc) | QQP (f1/acc) | RTE (acc) | SST-2 (acc) | STS-B (Pear./Spear.) | Avg. |
|---|---|---|---|---|---|---|---|---|---|---|
| RoBERTa | 32-32-32 | 62.50 | 87.75/87.23 | 93.1/90.44 | 92.68 | 88.78/91.6 | 80.51 | 95.18 | 91.04/90.72 | 86.40 |
| Quant-Noise [47] | PQ | - | 83.60/- | - | - | - | - | - | - | - |
| PACT [41] | 4-4-4 | 19.43 | 78.72/79.55 | 81.42/73.04 | 84.55 | 85.14/88.91 | 58.12 | 88.76 | 72.15/72.46 | 70.82 |
| LSQ+ [12] | 4-4-4 | 24.69 | 83.28/83.24 | 83.17/75.0 | 85.12 | 86.96/90.22 | **58.12** | 89.79 | 78.08/78.41 | 73.36 |
| **Ours** | 4-4-4 | **37.10** | **84.91/85.2** | **84.60/77.70** | **87.66** | **87.24/90.52** | 57.76 | **90.25** | **85.61/85.33** | **76.67** |
| LSQ+(+KD) | 4-4-4 | 30.33 | 87.17/87.27 | 89.39/85.05 | 91.87 | 88.56/91.48 | 61.73 | 92.20 | 83.18/83.10 | 77.97 |
| **Ours(+KD)** | 4-4-4 | **48.78** | **87.33/87.16** | **91.92/88.97** | **91.93** | **88.81/91.67** | **66.79** | **92.43** | **88.97/88.76** | **82.09** |
| BART | 32-32-32 | 56.32 | 86.45/86.55 | 91.37/87.5 | 92.31 | 88.34/91.39 | 79.06 | 93.35 | 90.11/89.94 | 84.61 |
| PACT | 4-4-4 | 18.72 | 80.57/80.36 | 87.99/82.60 | 85.52 | 85.09/88.19 | 57.40 | 89.45 | 87.49/87.36 | 73.86 |
| LSQ+ | 4-4-4 | 18.12 | 82.41/82.29 | 88.35/83.58 | 87.39 | 86.04/89.64 | 57.40 | 90.48 | 86.89/86.86 | 74.55 |
| **Ours** | 4-4-4 | **50.83** | **84.81/84.57** | **90.94/87.01** | **90.92** | **87.88/90.93** | **73.29** | **92.43** | **89.22/89.02** | **82.46** |

Table 16: Comparison among different QAT strategies with low-bit activation on GLUE benchmark for RoBERTa and BART.

| Tasks | GLUE | | | XSum |
|---|---|---|---|---|
| Bits(W-E-A) | BERT | RoBERTa | BART | BART |
| 32-32-32 | 417.6 | 475.5 | 534.1 | 531.8 |
| 8-8-8 | 104.8 | 119.2 | 134.0 | 133.4 |
| 6-6-6 | 78.7 | 89.5 | 100.6 | 100.2 |
| 4-4-8 | 52.6 | 59.8 | 67.3 | 67.0 |
| 4-4-4 | 52.6 | 59.8 | 67.3 | 67.0 |
| 2-2-4 | 26.5 | 30.1 | 34.0 | 33.8 |

Table 17: Model size(MB) of quantized models.

---

**Algorithm 1:** Token-Wise Clipping

**Input:** grid search iteration $K$, model with $L$ layers, number of tokens $T$.

{1. Coarse stage:}

$loss = \text{INF}, s_0 = 1.0$

**for** $k = 0$ *to* $K - 1$ **do**

    $\alpha = 1 - 0.01 * k$;

    **for** $i = 1$ *to* $L$ **do**

        layer input $\boldsymbol{X}$, token $t$ at embedding $j$ $\boldsymbol{X}_{t,j}$;

        $\boldsymbol{o}^u = \{\max_j \boldsymbol{X}_{1,j}, \max_j \boldsymbol{X}_{2,j}, ..., \max_j \boldsymbol{X}_{T,j}\}$;

        $\boldsymbol{o}^l = \{\min_j \boldsymbol{X}_{1,j}, \min_j \boldsymbol{X}_{2,j}, ..., \min_j \boldsymbol{X}_{T,j}\}$;

        $c^u = quantile(\boldsymbol{o}^u, \alpha), \ c^l = quantile(\boldsymbol{o}^l, \alpha)$;

        $\boldsymbol{X} = clip(\boldsymbol{X}, c^l, c^u)$;

    Calculate step size $s$ and quantization loss Eq. (6);

    **if** $loss > loss_k$ **then**

        $loss = loss_k, s_0 = s$ ;

Find the initialization step size $s_0$;

{2. Fine-grained stage:}

Optimize the $s$ using Eq. (9) with Eq. (6);

**return** Optimized step size $s$ ;