# OpenReview forum: "Outlier Suppression: Pushing the Limit of Low-bit Transformer Language Models"
_NeurIPS.cc/2022/Conference — NeurIPS 2022 Accept_

### Official Review · Reviewer_GEPe · 2022-07-06

**Rating:** 7
**Confidence:** 4
**Soundness:** 3 good
**Presentation:** 3 good
**Contribution:** 3 good

**Summary:**

This manuscript focuses on Transformer quantization for Bert. It first analyzes the performance regression from outliers during model quantization, specifically in LayerNorm. Following this, the authors propose a Gamma Migration approach to change how we consider the parameter Gamma in LayerNorm, during quantization. In addition, they propose a coarse-to-fine algorithm to find the outliers during token-wise clipping.

**Questions:**

* What is the total number of parameters of the models in the experiments? With a fixed size training set, the degradation caused by quantization should be highly related to the number of parameters. Please consider adding these numbers to the tables. If possible, please also conduct experiments on different model sizes to see if the proposed approach is beneficial to Transformer models in GLUE in general.
* For PTQ, what would be the performance of 4-bit quantization? I understand that 4-4-4 PTQ with any quantization paradigm might resulting in garbage results. However, it would interesting to see something like 4-8-8 or 4-6-6. The rationale is that, in real-world scenarios like on-device models, memory usage could be the bottleneck, while the inference speed with 8bit or 6bit are fast enough. In this case, 4bit weight quantization would be very helpful in reducing memory usage.


**Limitations:**

The authors mentioned that "In Discussions we leave some topics as future work." However, I would suggest to explicitly summarize them either in a separate discussion section or change conclusions to something like "conclusions and discussions of limitations". Also, I don't think the discussions in the results section covered "the limitations and potential negative societal impact of their work" well enough. Please revise according to NeurIPS requirements.


**Strengths And Weaknesses:**

Strengths
* The paper is well organized and very easy to follow
* The references are enough
* The technical novelty is enough (see summary)

Weakness
* Although the authors conducted extensive experiments, there are still some key aspects in results missing. See below

---

> ### Author Response · Authors · 2022-08-02
> **Response to Reviewer GEPe Part II**
>
> - Q2: For PTQ, what would be the performance of 4-bit quantization? I understand that 4-4-4 PTQ with any quantization paradigm might result in garbage results. However, it would interesting to see something like 4-8-8 or 4-6-6. The rationale is that, in real-world scenarios like on-device models, memory usage could be the bottleneck, while the inference speed with 8bit or 6bit is fast enough. In this case, 4bit weight quantization would be very helpful in reducing memory usage.
>
>   A: It is indeed interesting to investigate the 4-bit weight quantization setting that helps reduce model size. We tried 4-6-6 bit on BERT-base and RoBERTa-base. The results in the two tables below show that our methods can also benefit the 4-bit quantization and is helpful with the device of constrained memory.
>
>   For BERT-base models, our outlier suppression framework achieves near-floating point performance with a reduction of 4.83% on these so small models with low-bits, while others suffer from a performance degradation of 15.4% and 13.74%, respectively.  For RoBERTa-base, though the results reveal that it’s hard to be close to FP values on 4-6-6 settings, ours still helps a lot and outperforms existing methods by about 15.62%.
>
>   | RoBERTa-base (W4E6A6) | Model Size (MB) | CoLA | MNLI | MRPC | QNLI | QQP | RTE | SST-2 | STS-B | Avg |
>   | --- | --- | --- | --- | --- | --- | --- | --- | --- | --- | --- |
>   |  |  | MR | acc m/mm | acc/f1 | acc | f1/acc | acc | acc | pearson/spearmanr |  |
>   | FP | 475.5 | 62.5 | 87.75/87.23 | 90.44/93.1 | 92.68 | 88.78/91.6 | 80.51 | 95.18 | 91.04/90.72 | 86.40 |
>   | OMSE | 69.1 | 0.0 | 55.48/57.33 | 68.38/75.14 | 61.12 | 81.57/86.3 | 50.9 | 81.42 | 38.47/37.58 | 55.45 |
>   | Percentile | 69.1 | 2.11 | 59.06/61.85 | 53.92/52.76 | 61.54 | 75.14/83.33 | 47.65 | 86.24 | 56.98/57.07 | 55.95 |
>   | Ours | 69.1 | 30.26 | 76.02/76.99 | 73.77/80.65 | 78.22 | 83.01/87.5 | 57.76 | 89.91 | 76.95/77.9 | 71.57 |
>
>   | BERT-base (W4E6A6) | Model Size (MB) | CoLA | MNLI | MRPC | QNLI | QQP | RTE | SST-2 | STS-B | Avg |
>   | --- | --- | --- | --- | --- | --- | --- | --- | --- | --- | --- |
>   |  |  | MR | acc m/mm | acc/f1 | acc | f1/acc | acc | acc | pearson/spearmanr |  |
>   | FP | 417.6 | 59.6 | 84.94/84.76 | 87.75/91.35 | 91.84 | 87.82/90.91 | 72.56 | 93.35 | 89.70/89.28 | 83.83 |
>    | OMSE | 58.3 | 29.79 | 69.47/68.76 | 75.74/81.43 | 78.66 | 67.19/78.36 | 62.82 | 84.86 | 83.93/84.29 | 70.09 |
>   | Percentile | 58.3 | 25.63 | 67.98/69.04 | 78.92/83.27 | 69.34 | 67.63/78.31 | 58.12 | 86.12 | 84.95/86.37 | 68.43 |
>   | Ours | 58.3 | 52.99 | 80.26/80.49 | 81.62/85.93 | 88.17 | 79.27/85.38 | 65.34 | 91.97 | 87.09/87.15 | 79.08 |
>
> - Q3: The authors mentioned that "In Discussions we leave some topics as future work." However, I would suggest to explicitly summarize them either in a separate discussion section or change conclusions to something like "conclusions and discussions of limitations". Also, I don't think the discussions in the results section covered "the limitations and potential negative societal impact of their work" well enough. Please revise according to NeurIPS requirements.
>
>   A: Thanks for pointing that out.  We have changed conclusions to “conclusions and discussions of limitations” and clarified the limitation and future work in the paper.
>
>   In this paper, we mainly analyze the challenge of language transformer quantization. It is valuable to systematically explore whether the conclusion in this paper benefits other fields such as computer vision. And as we mention in the Appendix that the outlier emergence involves not only the fine-tuned models but also the pre-trained ones. Diving into the pre-training process is also a profound future topic for a better understanding of the outliers.

---

> > ### Comment · Reviewer_GEPe · 2022-08-08
> > **Thanks for the response**
> >
> > Thanks for the responses to my questions. The explanations and the experiments added answered my questions well, which makes this manuscript more solid. Increased my scores.

---

> ### Author Response · Authors · 2022-08-02
> **Response to Reviewer GEPe Part I**
>
> We would like to sincerely thank the reviewer for providing insightful suggestions on this paper. We have revised the paper and added the necessary experiments as suggested. The detailed response is listed below. We hope our reply can address the questions.
>
> - Q1: What is the total number of parameters of the models in the experiments? With a fixed size training set, the degradation caused by quantization should be highly related to the number of parameters. Please consider adding these numbers to the tables. If possible, please also conduct experiments on different model sizes to see if the proposed approach is beneficial to Transformer models in GLUE in general.
>
>   A: Thanks for your constructive advice on presenting the model size, and we have added these numbers to the tables.
>
>   Apart from the 6-bit RoBERTa-base model (89.5MB) in the original submission, we also conducted experiments on 6-bit DistilRoBERTa (58.9MB) and RoBERTa-large (255.2MB) to explore the performance of models with different sizes. On RoBERTa-base model, we have achieved the 8.64% average enhancement. And the performance on DistilRoBERTa and RoBERT-large can also be boosted by 7.36%, and 2.59% respectively.
>
>   Detailed results are listed in the following two tables. For 6-bit DistilRoBERTa with a much smaller size, our methods outperform others by a large margin. For RoBERTa-large, ours also shows satisfying outcomes consistently, while Percentile behaves much worse on STS-B, QQP and OMSE do not work well on MRPC and QNLI. Moreover, we will cover more validations on other models in the future.
>
>   | DistilRoBERTa (W6E6A6)  | Model Size (MB) | CoLA | MNLI | MRPC | QNLI | QQP | RTE | SST-2 | STS-B | Avg |
>   | --- | --- | --- | --- | --- | --- | --- | --- | --- | --- | --- |
>   |  |  | MR | acc m/mm | acc/f1 | acc | f1/acc | acc | acc | pearson/spearmanr |  |
>   | FP | 313.3 | 60.77 | 84.10/84.38 | 87.50/91.28 | 91.07 | 87.32/90.56 | 71.84 | 92.20 | 88.55/88.21 | 83.35 |
>   | OMSE | 58.9 | 8.33 | 81.62/81.49 | 75.0/83.65 | 80.98 | 82.49/85.84 | 65.7 | 89.11 | 78.18/78.07 | 70.91 |
>   | Percentile | 58.9 | 28.47 | 78.45/78.71 | 75.25/84.49 | 84.18 | 79.16/81.64 | 58.12 | 89.68 | 75.5/76.44 | 71.91 |
>   | Ours | 58.9 | 47.86 | 82.93/82.41 | 80.88/87.09 | 88.32 | 85.12/89.09 | 67.51 | 91.86 | 84.99/84.78 | 79.27 |
>
>   | RoBERTa-large (W6E6A6) | Model Size (MB) | CoLA | MNLI | MRPC | QNLI | QQP | RTE | SST-2 | STS-B | Avg |
>   | --- | --- | --- | --- | --- | --- | --- | --- | --- | --- | --- |
>   |  |  | MR | acc m /mm | acc/f1 | acc | f1/acc | acc | acc | pearson/spearmanr |  |
>   | FP | 1355.6 | 67.74 | 90.16/90.06 | 87.99/91.27 | 94.69 | 89.58/92.14 | 84.84 | 96.33 | 91.82/91.70 | 88.25 |
>   | OMSE | 255.2 | 56.20 | 85.83/85.38 | 75.0/84.45 | 86.07 | 85.4/89.19 | 72.57 | 93.92 | 85.49/85.54 | 80.86 |
>   | Percentile | 255.2 | 55.73 | 84.95/85.37 | 81.86/87.33 | 90.92 | 79.21/85.92 | 70.04 | 93.0 | 82.16/82.37 | 80.53 |
>   | Ours | 255.2 | 58.52 | 85.16/85.52 | 83.58/89.0 | 91.05 | 85.7/89.49 | 78.34 | 94.27 | 86.17/86.22 | 83.45 |

---

### Official Review · Reviewer_LNdx · 2022-07-10

**Rating:** 7
**Confidence:** 3
**Soundness:** 3 good
**Presentation:** 4 excellent
**Contribution:** 4 excellent

**Summary:**

The papers pursues a better solution to deal with outliers in special tokens that are considered as the critical bottleneck for the quantization accuracy. They identify that it is the \gamma variable in LayerNorm which amplifies outliers and propose an outlier suppression framework, consisting of Gamma Migration and Token-wise Clipping to overcome the quantization bottleneck. Extensive results show that the proposed framework outperforms existing quantization algorithms and for the time time fully recovers full-precision BERT results with a  6-bit post training quantization model and 4-bit model produced by quantization-aware training.

**Questions:**

- Missing citation: [Compression of Generative Pre-trained Language Models via Quantization](https://arxiv.org/pdf/2203.10705.pdf(https://arxiv.org/pdf/2203.10705.pdf), they developed a dynamic scaling approach and it would be nice to understand how are the observations in two papers are related with each other.
- Figure 2 is cited in texts multiple times but is not present in the paper.


**Limitations:**

Yes.

**Strengths And Weaknesses:**

### Strengths
- The proposed approach is well motivated by the observation of the layernorm outliers and the experiments are extensive and empirically strong on a wide range of tasks including sentence classification, question answering, and summarization.
- The analysis and visualization are very informative and well presented

### Weaknesses
- One figure is missing from the description
- Some results could be better justified. e.g., It seems that Gamma Migration and Token-Wise Clipping complement each other, but why? How to understand the joint effect of the two techniques and how do they compare to approaches like QBERT or Q8BERT?

---

> ### Author Response · Authors · 2022-08-02
> **Response to Reviewer LNdx Part II**
>
> - Q3: Missing citation: [Compression of Generative Pre-trained Language Models via Quantization](https://arxiv.org/pdf/2203.10705.pdf(https://arxiv.org/pdf/2203.10705.pdf), they developed a dynamic scaling approach and it would be nice to understand how are the observations in two papers are related with each other.
>
>   A: Thanks for pointing out the relevant paper. We have cited it in the revision and added analyses of the relationship between these two papers.
>
>   We find that the two papers have different observations and different methods. And both two works are devoted to the quantization bottleneck of language models. They notice that due to the sequential computation nature of generative models, word embedding is easier to be homogeneous, and propose a token-level contrastive distillation method. They also observe the outliers in weights and suggest a dynamic scaling technique, which calculates a good clipping range for weights during QAT. Our work begins from the activation outlier phenomenon and correlates it with obvious performance degradation. Concerning the structured characteristic of outliers, we propose Gamma Migration to weaken the outliers from the origin, and the Token-Wise Clipping calculates a good clipping range for activation. They are applied to the calibration phase, thus both suits PTQ and QAT with a better initialization point.
>
>   To summarize, our work and theirs both work well and target different parts of quantization on language models. They focus on the inherent quantization bottleneck for weights for generative models. We pay much attention to the structured activation outliers of language models, including classification and generative aspects.
>
> - Q4: One figure is missing from the description. Figure 2 is cited in texts multiple times but is not present in the paper.
>
>   A: Thanks for pointing that out, and sorry for the confusion caused by an error from latex autoref. The “Figure 2” mentioned in the text should be “Figure 1”. We have fixed this error in the revision.

---

> ### Author Response · Authors · 2022-08-02
> **Response to Reviewer LNdx Part I**
>
> Thanks for the reviewer’s valuable suggestions and positive feedback on this paper. Below is the detailed response to each question. Hope the following reply helps address the concerns.
>
> - Q1: Some results could be better justified. e.g., It seems that Gamma Migration and Token-Wise Clipping complement each other, but why? How to understand the joint effect of the two techniques?
>
>   A: Thanks for the suggestion. The joint improvement brought by Gamma Migration and Token-Wise Clipping is mainly because they suppress the outliers from different perspectives. Here we give a detailed explanation below.
>
>   Gamma Migration eliminates the outlier amplification phenomenon in LayerNorm and weakens the outliers. After getting rid of this outlier amplification, some specific tokens still suffer from an outlier problem which is harmful for the quantization. Thus Token-Wise Clipping is further devised to suppress these outliers by appropriately clipping them. It takes advantage of the outliers' importance and tokens' information and thus obtains a better trade-off between clipping error and rounding error compared with other calibration algorithms.
>
>   In other words, Token-Wise Clipping acts as a better quantization calibration algorithm that finds a better clipping range, and Gamma Migration acts as a layer transformation plugin that removes the outlier amplifier without extra computation overhead. In theory, Gamma Migration can be combined with any calibration algorithms for improvement, and in practice, due to the superiority of Token-Wise Clipping, the combination of these two methods achieves the best performance.
>
>   From the above analyses, we can find that these two techniques solve different parts of problems and thus complement each other. Benefitting from the power of each method, combining them achieves the best results and thereby pushes the limit of low-bit language transformers to a new SOTA. We are sorry for the confusion and have revised the paper to make these points clearer.
>
> - Q2: Some results could be better justified. e.g., how do Gamma Migration and Token-Wise Clipping compare to approaches like QBERT or Q8BERT?
>
>   A: Thanks for the useful advice. Here, we explain the Q-BERT, Q8BERT, and our outlier suppression framework from technique design and experimental results to help the understanding of their relationships.
>
>     - For technique design, both of the two mentioned methods are quantization-aware training (QAT) algorithms. Q8BERT shows the scheme to perform QAT into the fine-tuning phase of BERT. Q-BERT advises a new group-wise quantization scheme and explores the mixed-precision quantization using Hessian information. Our framework investigates the structured outliers, which hurts the quantization performance seriously, and devises two methods to eliminate the outlier amplification effect and find a suitable clipping range. The framework both suits post-training quantization (PTQ) and quantization-aware training. In QAT, our methods apply at the calibration phase and provide good initialization for later training.
>     - For experimental results, as our framework is devoted to the calibration phase of QAT, we select a strong quantization training baseline LSQ+ and combine our methods with it. In the original paper, we listed the results of different QAT methods, including Q-BERT, Q8BERT and LSQ+. It can be seen that the LSQ+ (4-4-8 bit) has surpassed Q-BERT (8-4-8 bit) by 7.8% and Q8BERT (8-8-8 bit) by 0.6% on average. And as a better initialization, our methods continue to improve the results upon LSQ+ by 1.26% on 4-4-8 bit (the LSQ+ on this setting has already pursued little accuracy degradation from the full-precision counterpart) and 9.64% on the 4-4-4 setting. This indicates that even combined with a strong baseline, our method can still bring an extra accuracy increase. Besides, Figure 5 in the paper demonstrates how a good initialization influences QAT training. In short, with a good starting point, the training converges faster, and the loss soon reaches a lower, more stable level.
>     - In fact, Q8BERT and Q-BERT target the training procedure or the quantization scheme, such as group-wise and mixed-precision schemes. Ours targets at the inherent harmful outliers of language transformer models. Thus, we work on different aspects of quantization. Moreover, as our framework involves the calibration phase in QAT, our methods can also be combined with theirs, such as applying to the mixed-precision problem. And we will validate this point in the future.
>
>   We have revised the discussion for this part in the paper to make the readers better understand the contribution of our methods.

---

### Official Review · Reviewer_U3Hv · 2022-07-11

**Rating:** 7
**Confidence:** 4
**Soundness:** 4 excellent
**Presentation:** 2 fair
**Contribution:** 3 good

**Summary:**

The paper tries to make improvements on the problem of quantizing transformers, wherein the key problem is the presence of outliers. The paper presents an analysis of the problem and tries to track the origin of outliers and find that the scaling parameter (gamma) in layer norm acts as an outlier amplifier. The paper then presents a method named Gamma Migration, which moves computations associated with the gamma parameter to subsequent layers. This method improves quantization performance against a simple minmax baseline. The key gains in the paper come from token-wise clipping, in which the authors propose coarse to fine grained pipeline to clip the more aggressive outliers.

**Questions:**

1. The ablation study points to the limited efficacy of gamma migration. Have you conducted an analysis of why is this the case? Given that gamma amplification is presented as the origin of the outlier problem, a conclusive analysis of its impact will be useful.
2. Have you tried the proposed methods (Gamma migration and Token-wise clipping) on GPT-2?
3. Do you have any comments on whether the work is generic to transformers or is highly applied on the text domain? For example, does the analysis hold on vision transformers or multimodal models?
4. The tasks tackled are only classification tasks, where word-similarity post quantization can give a good estimate of downstream performance. Have you evaluated any generation tasks on BART?

**Ethics Review Area:**

["I don’t know"]

**Limitations:**

The authors haven't adequately addressed the limitations and the pain points on reproducing this work. The negative societal impact is irrelevant for this work.

**Strengths And Weaknesses:**

The strengths of the paper are:

1. The empirical results are considerably strong. The 6-bit PTQ results are the first such high-quality results on GLUE.
2. Given the importance of pre-trained transformer LMs such as BERT, Roberta, the proposed methods can help system developers and deployers.
3. Each of the methods (although highly applied) is well motivated, with good intermediate quantifications to illustrate their utility.

The paper has the following weaknesses:

1. In general, the paper writing can certainly be improved, e.g. a number of acronyms such as PTQ, QAT are never introduced to the reader.
2. Limited novelty of the work. The ablation study points out that token clipping is the most important component, and that gamma migration is of limited importance. This does not gel well with the fact that the analysis on the origin of the outliers is presented as a significant work.

---

> ### Author Response · Authors · 2022-08-02
> **Response to Reviewer U3Hv Part IV**
>
> * Q4: The tasks tackled are only classification tasks, where word-similarity post quantization can give a good estimate of downstream performance. Have you evaluated any generation tasks on BART?
>
>   A: We highly agree that it is meaningful to evaluate both the classification and generation tasks. Thus in the initial submission, we have also conducted experiments on the generation tasks (XSUM and CNN/DailyMail) using BART. As shown in Table 7 in the paper, our methods can achieve an improvement of 3%-4% for the 6-bit setting. This proves that our analyses also stand under the generation task setting and the proposed outlier suppression framework is general.
>
>   As for the word-similarity post quantization mentioned by the reviewer, we guess it might refer to the fact that word similarity is related to the effect of classification to some extent but can not reflect the effect of generation. Although these two tasks have different objectives, as long as we can reduce the quantization error and align the outputs of the quantized model to the full-precision model as close as possible, the quantization accuracy can naturally be improved. The experimental results verify this point.
>
>   For a more intuitive understanding, we show some generated sequences produced by the origin full-precision model, our quantized model, and trivial quantized model, respectively. From the table below, we can find that the summary generated by our quantized model is much closer to the full-precision one while the trivial one collapses.
>    | Article    | Following last year's successful U.K. tour, Prince and 3rdEyeGirl are bringing the Hit & Run Tour to the U.S. for the first time. The first -- and so far only -- scheduled show will take place in Louisville, Kentucky, the hometown of 3rdEyeGirl drummer Hannah Welton. Slated for March 14, tickets will go on sale Monday, March 9 at 10 a.m. local time. Prince crowns dual rock charts . A venue has yet to be announced. When the Hit & Run worked its way through the U.K. in 2014, concert venues were revealed via Twitter prior to each show. Portions of the ticket sales will be donated to various Louisville charities. See the original story at Billboard.com. \u00a92015 Billboard. All Rights Reserved. |
>   | ---------- | ------------------------------------------------------------ |
>   | FP         | Prince and 3rdEyeGirl are bringing the Hit & Run Tour to the U.S. for the first time. The first -- and so far only -- scheduled show will take place in Louisville, Kentucky. Portions of the ticket sales will be donated to various Louisville charities. |
>   | Percentile | The first -- and so far only only -- scheduled show will take the Hit & Run to the U U.U.S. following last year's successful U.K. tour. Prince and 3rd3rd3's hit hit hit the hit & Run is bringing the Hit and Run Tour to the United States for the first time. The first - and the so far far only scheduled shows will take place in Louisville, Kentucky. |
>   | Ours       | Prince and 3rdEyeGirl are bringing the Hit & Run Tour to the U.S. for the first time. The first -- and so far only -- scheduled show will take place in Louisville, Kentucky. Portions of the ticket sales will be donated to various Louisville charities. |
> * Q5: In general, the paper writing can certainly be improved, e.g. a number of acronyms such as PTQ, QAT are never introduced to the reader.
>
>   A: Thanks for pointing that out. We have revised our paper and introduced the acronyms in a suitable position to make the paper easier to follow.

---

> > ### Comment · Reviewer_U3Hv · 2022-08-08
> > **Improved Soundness Score**
> >
> > I appreciate the detailed responses and the additional experiments conducted to answer my questions. I have increased the soundness score for the paper.

---

> ### Author Response · Authors · 2022-08-02
> **Response to Reviewer U3Hv Part III**
>
> * Q3: Do you have any comments on whether the work is generic to transformers or is highly applied on the text domain? For example, does the analysis hold on vision transformers or multimodal models?
>
>   A: In this paper, we mainly focus on the language task, and it is also interesting and valuable to investigate the phenomenon of vision and multimodal tasks. In the limited time, we analyze the outlier phenomenon, apply our methods to vision transformers, and find that our framework can also help with quantization in the vision domain. For multimodal models, we will further analyze and validate them in the future. The detailed analyses and effects of vision transformers are given below.
>
>   * For Gamma Migration, we observe that there are also outliers that emerge in the LayerNorm’s output for vision transformers. Different from language models, outliers at some embedding dimensions are amplified by the scaling parameter at the same dimension in the LayerNorm, and outliers at some other dimensions are alleviated by the corresponding scaling parameter. Therefore, we migrate the scaling parameters that amplify the outliers into the weights of later layer and keep others still in the LayerNorm. Naturally, this can reduce the quantization error by eliminating the amplification effect.
>
>      As for the computation overhead, fortunately, these models take the pre-norm where the layer normalization is put inside the residual connection. Then, we do not need to consider transferring the amplifier into two branches (weight in the next layer and shortcut) like post-norm, but just let the weight absorb it. Thus not transferring the whole scaling parameters will not increase any computation costs during inference.
>
>   * For Token-Wise Clipping, we observe the [CLS] token in vision transformers often holds more aggressive outliers, which is similar to the phenomenon in language models. As for other tokens, we notice the meaning of tokens is indeed not the same as language tasks. In language tasks, tokens are usually fixed from the beginning and we have a vocabulary file. In vision tasks, tokens represent the information of patches in an image, which covers a large number of combinations of pixels and the information is often caught by a convolution module first. Therefore, more in-depth investigations about tokens in vision models need to be made in the future. Right now, we directly implement the Token-Wise Clipping in the vision tasks and surprisingly get good results. It reveals that there are also unimportant outliers in vision transformers, and Token-Wise Clipping can effectively find them.
>
>   * To give a concrete example, we conduct experiments on DeiT-Base model with patches of 16 x 16 size, resolution of 224 x 224 size with ImageNet dataset. The model’s full-precision performance is 81.80, and we take 6-6-6 bit setting for quantization.  The following table shows the effect of each part, which demonstrates the generalization ability of our methods.
>
>     |                     | MinMax    | Percentile | Token-Wise Clipping |
>     | ------------------- | --------- | ---------- | ------------------- |
>     | w/o Gamma Migration | 78.04     | 78.81      | 79.76               |
>     | w/ Gamma Migration   | **79.05** | **80.11**  | **81.12**           |
>
>     Besides comparison with common calibration algorithms for post-training quantization, we also consider some recent post-training quantization works likePTQ4ViT[1] and PTQ-ViT [2] on vision transformers. The table below proves our superiority in quantization by suppressing the outliers.
>
>     |  | DeiT-B/224 |
>     | --- | --- |
>     | FP  | 81.80 |
>     | PTQ4ViT [1] | 80.25 |
>     | PTQ-ViT [2] | 77.02|
>     | Ours | **81.12** |
>
>     [1] Yuan Z, Xue C, Chen Y, et al. PTQ4ViT: Post-Training Quantization Framework for Vision Transformers[J]. arXiv preprint arXiv:2111.12293, 2021.
>
>     [2] Liu Z, Wang Y, Han K, et al. Post-training quantization for vision transformer[J]. Advances in Neural Information Processing Systems, 2021, 34: 28092-28103.

---

> ### Author Response · Authors · 2022-08-02
> **Response to Reviewer U3Hv Part II**
>
> * Q2: Have you tried the proposed methods (Gamma migration and Token-wise clipping) on GPT-2?
>
>   A: Thanks for the suggestion. It is of high value to investigate the effect on more types of language models. We validated the effect of the proposed methods for GPT-2 on the WikiText 103 dataset for 8-bit quantization. The results are listed below. It can be seen that for GPT-2, our methods also achieve a consistent improvement compared with others, proving its effectiveness and generalization. We will add more comprehensive verification on other datasets in the future.
>
>   |            | WikiText 103 |
>   | --- | --- |
>   | GPT-2      |              |
>   | FP         | 15.97        |
>   | MinMax     | 25.32        |
>   | Percentile | 23.17        |
>   | OMSE       | 21.09        |
>   | Ours       | **17.16**        |
>
>   To explain the results, we observe that the GPT-2 model also suffers from the outlier amplification phenomenon due to the scaling parameter of LayerNorm.  And concerning the outlier importance is also beneficial for finding a clipping range. Therefore, our methods achieve better PPL.

---

> ### Author Response · Authors · 2022-08-02
> **Response to Reviewer U3Hv Part I**
>
> We would like to thank the reviewer for the valuable suggestions and thoughtful insight on this paper. The detailed response is listed below.  Hope our reply can address the concerns.
>
> * Q1: The ablation study points to the limited efficacy of gamma migration. Have you conducted an analysis of why is this the case? Given that gamma amplification is presented as the origin of the outlier problem, a conclusive analysis of its impact will be useful. (The ablation study points out that token clipping is the most important component, and that gamma migration is of limited importance. This does not gel well with the fact that the analysis on the origin of the outliers is presented as a significant work.)
>
>   A: Sorry for the confusion. Here we make a further clarification for a better understanding of how Gamma Migration and Token-Wise Clipping help the outlier suppression.
>
>   * First, for the phenomenon of outliers, we want to indicate that **(1) there have existed outliers in some embedding dimensions, especially for some specific tokens, and (2) the gamma in LayerNorm further amplifies them, making the quantization more intractable.** To suppress these harmful outliers for quantization, both problems should be handled.
>
>     **For the first problem**, we think it might correlate with token frequency during the pre-training phase (see Appendix C.2). In this paper, we aim to improve the quantization accuracy with as low costs as possible. Thus it is unrealistic to adjust the pre-training phase. But the observations that some outliers are unimportant inspire us to pursue a suitable clipping range for a good trade-off between clipping error and rounding error for quantization. Therefore, the Token-Wise Clipping is proposed to clip the outliers directly and appropriately under limited bit-width. Compared to previous calibration algorithms like MinMax, OMSE, and Percentile, which also aim to find a clipping range for quantization. Our method considers outlier importance and leverages the token’s characteristics thus works more effectively and efficiently.
>
>     **For the second problem**, Gamma Migration transforms gamma into the later layer and eliminates the amplification effect from the origin. Then it contributes to a more quantization-friendly distribution without any extra inference time. Moreover, as a general module transformation technique, it can be combined with any common calibration methods and boost their performance by weakening the outliers in advance.
>
>     Therefore, both methods are important because they suppress the outliers from different parts.
>
>   * With the above explanation, we can have a better understanding of the ablation study results. Compared with MinMax, Token-Wise Clipping is a better calibration scheme since it can efficiently find the outliers and suitably clip them. Due to the page limit, the comparison experiments with other calibration methods were put in Appendix D.3 in the original paper. The Gamma Migration, as a general plug-in technique, have helped both MinMax and Token-Wise Clipping pursue better accuracy. It is designed to cooperate with calibration algorithms as a general and supplementary technique rather than replacing them. We also give an experiment based on the Percentile calibration algorithm to further verify this point. As shown in the following table, Gamma Migration helps Percentile achieve an accuracy enhancement of 2%-11% on the GLUE dataset. It can be seen that combined with Gamma Migration, the calibration algorithms (MinMax, Percentile, Token-Wise Clipping etc.) can enjoy further improvement.
>
>     | RoBERTa-base | bit  | CoLA | MNLI | MRPC | QNLI | QQP | RTE | SST-2 | STS-B | Avg |
>     | --- | --- | --- | --- | --- | --- | --- | --- | --- | --- | --- |
>     |  | W-E-A | MR | acc m /mm | acc/f1 | acc | f1/acc | acc | acc | Pear./Spear. |  |
>     | FP | 32-32-32 | 62.5 | 87.75/87.23 | 90.44/93.1 | 92.68 | 88.78/91.6 | 80.51 | 95.18 | 91.04/90.72 | 86.40 |
>     | Percentile | 6-6-6 | 20.73 | 72.23/73.68 | 78.43/84.83 | 77.16 | 82.21/87.44 | 62.82 | 88.19 | 79.41/79.64 | 70.98 |
>     | Percentile + Gamma Migration | 6-6-6 | 29.06 | 83.17/83.51 | 82.84/87.97 | 81.13 | 85.13/88.86 | 64.62 | 91.4 | 83.53/85.53 | 75.81 |
>
>     Therefore, by putting the Gamma Migration and Token-Wise Clipping together, the performance can be largely improved with outliers first weakened, then clipped appropriately. These two methods complement each other and jointly push the limit of low-bit quantization for language transformers.
>
>   Thanks for the valuable question and we have revised the paper to make these points clearer.

---

### Meta-Review · Area_Chair_KeSh · 2022-08-20

**Recommendation:** Accept
**Confidence:** Certain

**Metareview:**

This paper proposes an outlier suppression method to improve transformer quantization. The method is derived based on careful analysis and thorough experiments demonstrate the efficacy of it. All reviewers agreed that this is a good paper. I recommend acceptance.

**Award:**

No

---

### Decision · Program_Chairs · 2022-09-14

Accept